# Unraveling the mechanism for paired electrocatalysis of organics with water as a feedstock

Ganceng Yang[1], Yanqing Jiao[1], Haijing Yan[1✉], Ying Xie[1], Chungui Tian[1], Aiping Wu[1], Yu Wang[1] & Honggang Fu[1✉]

Paired electroreduction and electrooxidation of organics with water as a feedstock to produce value-added chemicals is meaningful. A comprehensive understanding of reaction mechanism is critical for the catalyst design and relative area development. Here, we have systematically studied the mechanism of the paired electroreduction and electrooxidation of organics on Fe-Mo-based phosphide heterojunctions. It is shown that active H* species for organic electroreduction originate from water. As for organic electrooxidation, among various oxygen species (OH*, OOH*, and O*), OH* free radicals derived from the first step of water dissociation are identified as active species. Furthermore, explicit reaction pathways and their paired advantages are proposed based on theoretical calculations. The paired electrolyzer powered by a solar cell shows a low voltage of 1.594 V at 100 mA cm$^{-2}$, faradaic efficiency of ≥99%, and remarkable cycle stability. This work provides a guide for sustainable synthesis of various value-added chemicals via paired electrocatalysis.

---

[1] Key Laboratory of Functional Inorganic Material Chemistry Ministry of Education of the People's Republic of China, Heilongjiang University, Harbin 150080, China. ✉email: yanhaijing@hlju.edu.cn; fuhg@hlju.edu.cn

Renewable energy utilization and synthesis of valuable building-block chemicals are important for future sustainable developments. Water is an abundant resource on earth. Water electrolysis coupled with renewable energy (solar, wind, etc.) has attracted intensive attention[1]. However, both hydrogen and oxygen evolution reactions (HER and OER) are inefficient due to the high activation barriers and particularly the sluggish kinetic of complex OER process hinders the water electrolysis efficiency[2,3]. Additionally, anodic product ($O_2$) is not a value-added chemical[4]. In contrast to $H_2$ and $O_2$ evolution, producing hydrogen and oxygen species from water before the generation of $H_2/O_2$ is more accessible, which are often active species in various catalytic reactions[5,6]. Upgrading cheap organics to valuable chemicals holds a crucial role in the chemical and pharmaceutical fields[7,8]. Traditional industrial approaches suffer from low selectivity/efficiency, expensive catalysts, and harsh conditions[9,10]. Given that, using hydrogen or oxygen species from renewable electricity driven water splitting to reduce or oxidize organics to produce value-added chemicals is a promising green route in terms of its low energy consumption, nonuse of chemical reductants/oxidants, and mild reaction conditions[11,12].

Some studies have reported the electrooxidation reactions (EOR) of biomass-derived organics by using water as oxygen source[13]. Thereinto, the electrooxidation of 5-hydroxymethylfurfural (5-HMF), obtained from the degradation of cellulose, to valuable 2,5-furandicarboxylic acid (2,5-FDCA) is the focus, which is a six electron-proton reaction[14]. Recently, it has been coupled with HER to elevate energy conversion efficiency[15,16]. Several compelling electrocatalytic materials (hp-Ni, CoP, $Ni_2P$, FeP-$MoO_2$, etc) have been reported[17–20]. Catalyzing electroreduction reactions (ERR) of organic by active hydrogen species from water at the cathode is equally important compared with $H_2$ production[21]. What's more, ERR is more suitable to pair with EOR than HER, in view of lower energy requirement, more matchable reaction rate and electron (or proton) numbers[22,23]. The nitro compounds (R-$NO_2$) exist in plants are reduced into amino derivatives (R-$NH_2$) with water as hydrogen source, which have been widely used in the field of medicine and fine chemical industry[24]. Taking the electroduction of 4-nitrobenzyl alcohol (4-NBA) as an example, the reduction product, 4-aminobenzyl alcohol (4-ABA), is a significant drug intermediate, for which the reduction process also undergoes a six electron-proton process[25,26]. Hence, pairing 4-NBA ERR with 5-HMF EOR is expected to synthesize two value-added chemicals at two sides with low energy consumption. Sun's group preliminarily investigated the paired electrocatalysis of organics with $NiB_x$ as the catalyst[27], but with relatively high energy consumption. Other derivative systems, such as cathodic electrocatalytic deuteration and its paired transformations with anodic amine or alcohol oxidation have also been studied recently[28,29]. However, the reaction mechanism including water activation and dissociation and hydrogen/oxygen participation patterns remains ambiguity so far. To expedite the advance of the nascent realm, pursuing more various low-cost and efficient electrocatalysts and clarifying the catalytic mechanism is highly desirable.

Biological nitroreduction can perform naturally with the aid of [FeMo]-nitrogenase[30–33]. Enlightened by this, Fe-Mo-based composites may be ideal catalyst candidates for 4-NBA ERR. Meanwhile, our previous work found that Fe-Mo-based composite presented a certain activity toward 5-HMF EOR, but the selectivity and efficiency still needed to be improved[20]. Ni is considered an ideal candidate, in view of the fact that the multivalent characteristic of Ni making it promising for organic oxidation[34,35]. So the introduction of Ni into the Mo-Fe system is expected to improve the selectivity and stability of 5-HMF EOR. Herein, we fabricated a FeP-MoP heterojunction grown on Fe foam (FeP-MoP/FF) for electroreduction of 4-NBA to 4-ABA.

For the convenience of industrial integration, by only introducing Ni (replacing FF by FeNi foam), a FeP-$NiMoP_2$ heterojunction on FeNi foam (FeP-$NiMoP_2$/FNF) was designed for electrooxidizing of 5-HMF to 2,5-FDCA. Due to little mutual interference between catalyst materials and matchable reaction kinetics of 4-NBA ERR and 5-HMF EOR, the paired electrolyzer powered by solar cell presents a low voltage of 1.594 V at 100 mA cm$^{-2}$, faradaic efficiency of ≥99% and remarkable cycle stability. More importantly, the reaction mechanisms of the paired electroreduction and electrooxidation system on Fe-Mo-based phosphide heterojunctions was systematically studied. Isotope labeling and $^1$H-NMR validate that H* species for 4-NBA electroreduction originate from water. As for 5-HMF electrooxidation, among various O species (OH*, OOH*, and O*), OH* free radicals from the first step of water dissociation are identified as the active species by performing electron spin resonance (ESR). The enhanced electronic conductivity and electron transfer in the FeP-MoP and FeP-$NiMoP_2$ heterojunctions optimize the of 4-NBA ERR and 5-HMF EOR reaction kinetics, contributing to the superior 4-NBA ERR and 5-HMF EOR activities. Furthermore, explicit reaction pathways on Fe-Mo-based phosphide heterojunctions and their paired advantages are proposed.

## Results

**Material synthesis and characterizations.** The FeP-MoP/FF and FeP-$NiMoP_2$/FNF electrodes used for solar-cell driven paired system were fabricated based on an in-situ growth route (Fig. 1a and Supplementary Fig. 1). For FeP-MoP/FF, Fe foam (FF) with rich porous structure and excellent conductivity was selected as both substrate and Fe source[36]. Ammonium heptamolybdate (APM) reacted with the in-situ released Fe species from FF by adding ascorbic acid, making in-situ growth of Fe-Mo oxide precursor with nanosheet morphology on FF under hydrothermal condition (Supplementary Fig. 2). Subsequently, FeP-MoP/FF was obtained by phosphorizating Fe-Mo oxide precursor at 600 °C. Similarly, the Fe-NiMo oxide precursor with wire-like morphology on FeNi foam (FNF) was prepared by replacing FF by FNF, (Supplementary Fig. 3). The phosphorization temperature was 500 °C for FeP-$NiMoP_2$/FNF. The decreased phosphorization temperature may be ascribed to the strong interaction of Ni with Mo, making Mo to be easily phosphorized[37]. Scanning electron microscopy (SEM) and transmission electron microscopy (TEM) images of FeP-MoP/FF reveal a coarse nanosheet with the size and thickness of 500 and 50 nm, respectively (Fig. 1b, c). Such sheet structure of FeP-MoP/FF enlarges the surface area and enables the contact with reactants, thereby promoting the catalytic performance. The high-resolution TEM (HRTEM) image (Fig. 1d) shows an obvious interface between FeP (211) (0.188 nm) and MoP (100) (0.278 nm), implying the formation of FeP-MoP heterojunction. Scanning transmission microscopy (STEM) image and the corresponding energy dispersive X-ray (EDX) mapping show homogeneous elemental distributions of Fe, Mo, and P in FeP-MoP/FF (Fig. 1e). SEM and TEM images of FeP-$NiMoP_2$/FNF show core-shell structural FeP-$NiMoP_2$ wires on the FNF substrate with the length and diameter of 5 μm and 150 nm. (Fig. 1f, g). A clear interface according to the lattice fringes of FeP (211) (0.188 nm) and $NiMoP_2$ (001) (0.288 nm) is observed (Fig. 1h), suggesting the generation of FeP-$NiMoP_2$. Furthermore, the Fe, Mo, Ni, and P elements are distributed homogeneously (Fig. 1i). More interestingly, Mo and Ni species are distributed in the inner core, while Fe and P species are obviously distributed in the outer shell, indicating that interaction between Mo and Ni is stronger than that between Mo and Fe. This is the reason that FeP and $NiMoP_2$ phases form easily in the heterojunction.

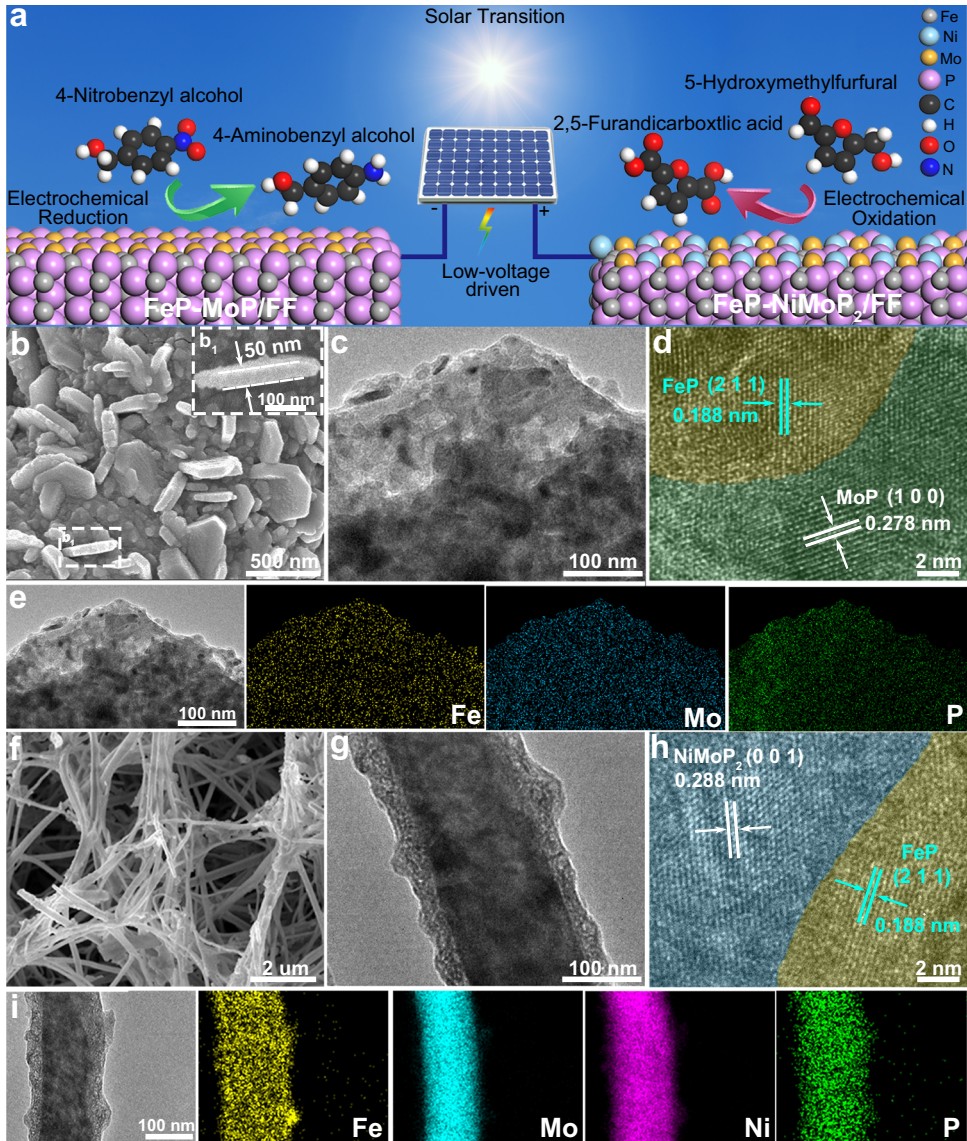

**Fig. 1 Morphology characterization of FeP-MoP/FF and FeP-NiMoP₂/FNF. a** Illustration for the paired 4-NBA ERR and 5-HMF EOR system driven by solar-cell. **b** SEM image (inset: the cross section SEM image), **c** TEM image, **d** HRTEM image. **e** STEM image and EDX mappings of Fe, Mo, and P of FeP-MoP/FF. **f** SEM image, **g** TEM image, **h** HRTEM image, **i** STEM image and EDX mappings of Fe, Mo, Ni, and P of FeP-NiMoP₂/FNF.

The structural investigation of FeP-MoP/FF and FeP-NiMoP₂/FNF was performed by X-ray diffraction (XRD) measurement. For FeP-MoP/FF (Fig. 2a), the diffraction peaks are well indexed to the metallic Fe (PDF No. 87-0722), monoclinic MoP (PDF No. 24-0771), and orthorhombic FeP (PDF No. 71-2262), respectively. Similarly, for FeP-NiMoP₂/FNF (Fig. 2b), the diffraction peaks can be indexed to alloy FeNi₃ (PDF No. 38-0419) from FNF, NiMoP₂ (PDF No. 33-0927), and FeP (PDF No. 71-2262), respectively. The above results demonstrate that FeP-MoP/FF and FeP-NiMoP₂/FNF were successfully fabricated. FeP/FF, MoP/FF, Ni₂P/NF, and NiMoP₂/NF samples were synthesized for comparison (Supplementary Figs. 4–7). X-ray photoelectron spectroscopy (XPS) was conducted to examine the elemental composition and interfacial interaction of FeP-MoP/FF and FeP-NiMoP₂/FNF. XPS survey spectra (Supplementary Fig. 8) provide solid evidence for the presence of Fe, Mo, P elements in the FeP-MoP/FF and FeP-NiMoP₂/FNF, and Ni element is only observed in the FeP-NiMoP₂/FNF. Fe 2p XPS spectrum (Fig. 2c) of FeP-MoP/FF can be deconvoluted into six sub-peaks, corresponding to Fe-P bond (706.3/719.1 eV), Fe-O bond (710.6/724.1 eV) and satellite peaks (714.8/728.8 eV), respectively[38]. Mo 3d XPS spectrum (Fig. 2d) of FeP-MoP/FF shows three doublets, which can be ascribed to Mo-P bond (229.0/232.1 eV), Mo$^{IV}$ species (230.6/233.7 eV) and Mo$^{VI}$ species (233.1/236.0 eV), respectively[39]. Fe 2p and Mo 3d spectra of FeP-NiMoP₂/FNF are similar to those of FeP-MoP/FF. Ni 2p XPS spectrum (Fig. 2e) of FeP-NiMoP₂/FNF displays three doublets, belonging to Ni-P bond (851.7/868.7 eV), Ni-O bond (855.2/871.8 eV) and satellite peaks (861.0/878.6 eV), respectively[40]. For P 2p XPS spectra (Fig. 2f) of FeP-MoP/FF and FeP-NiMoP₂/FNF, three sub-peaks of 128.8, 129.5, and 132.9 eV can be indexed to P 2p$_{3/2}$, P 2p$_{1/2}$, and P-O bond, respectively[41]. The above results further validate the successful formation of FeP-MoP/FF and FeP-NiMoP₂/FNF. Notably, Fe binding energy (BE) in FeP-MoP/FF negatively shift (0.3 eV) compared to FeP/FF. Meanwhile, a positive shift of 0.2 eV is observed for Mo BE in FeP-MoP/FF relative to those of MoP/FF. These observations demonstrate the electron transfer is from Mo to Fe, leading to electron-rich Fe and electron-deficient Mo. Such electron-rich Fe is similar to that of nitroreductase[30,42], which is conducive to 4-NBA ERR. As for FeP-NiMoP₂/FNF, the BEs of Mo and Fe elements show positive

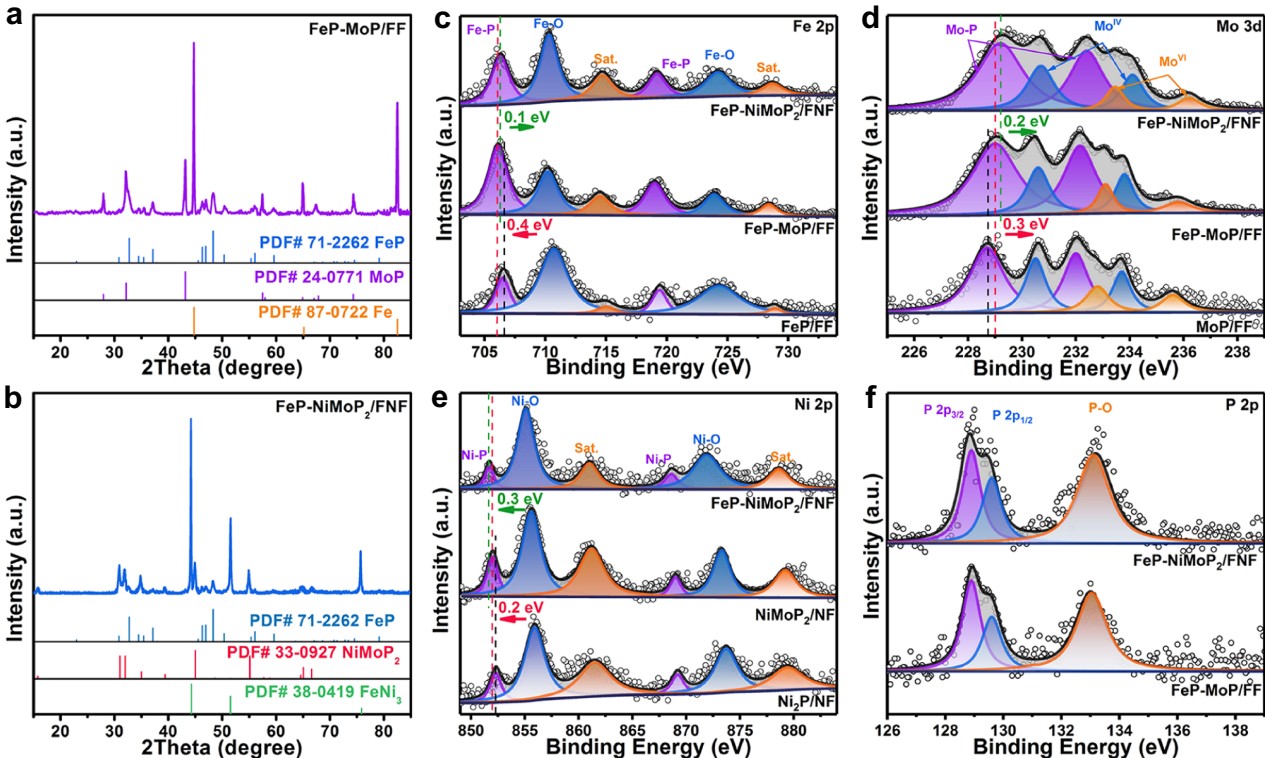

**Fig. 2 Structural characterizations of FeP-MoP/FF and FeP-NiMoP$_2$/FNF.** XRD patterns of (**a**) FeP-MoP/FF and **b** FeP-NiMoP$_2$/FNF. **c** Fe 2p XPS spectra of FeP/FF, FeP-MoP/FF and FeP-NiMoP$_2$/FNF. **d** Mo 3d XPS spectra of MoP/FF, FeP-MoP/FF and FeP-NiMoP$_2$/FNF. **e** Ni 2p XPS spectra of Ni$_2$P/NF, NiMoP$_2$/NF and FeP-NiMoP$_2$/FNF. **f** P 2p XPS spectra of FeP-MoP/FF and FeP-NiMoP$_2$/FNF.

shifts in comparison with those of FeP-MoP/FF, suggesting that both Mo and Fe lose electrons. With regard to Ni BE, a negative shift of 0.3 eV is observed in FeP-NiMoP$_2$/FF when contrasted to NiMoP$_2$/NF, and the Ni BE in NiMoP$_2$/NF shift 0.2 eV to lower BE compared with those of Ni$_2$P/NF. These results manifest that Ni has more strong capturing electron ability than Fe and it can captures electrons from both Fe and Mo, inducing the electron redistribution at the interface and resulting in more electron-deficiency on Mo in FeP-NiMoP$_2$/FNF, which may facilitate the improvement of 5-HMF EOR performance.

**The 4-NBA ERR performance.** The 4-NBA ERR performance of FeP-MoP/FF was studied by a typical three-electrode configuration in 1 M KOH with 10 mM 4-NBA, where the synthesized catalysts, Hg/HgO electrode and carbon rod were used as working electrode, reference electrode and counter electrode, respectively. The 4-NBA ERR activity of FeP-MoP/FF and FeP-NiMoP$_2$/FNF was first evaluated, where FeP-MoP/FF exhibits a better 4-NBA ERR activity than FeP-NiMoP$_2$/FNF (Supplementary Fig. 9). Therefore, we further studied the activity of FeP-MoP/FF toward 4-NBA ERR in detail. Figure 3a shows the linear sweep voltammetry (LSV) curves of FeP-MoP/FF for 4-NBA ERR (with 4-NBA) and HER (without 4-NBA). The potentials to achieve 10 and 100 mA cm$^{-2}$ ($\eta_{10}$ and $\eta_{100}$) for 4-NBA ERR on FeP-MoP/FF are 0.364 and 0.169 V vs. reversible hydrogen electrode (RHE)[43,44], which are 441 and 370 mV lower than those for HER (Fig. 3a and Supplementary Table 1). In the meantime, the Tafel slope for 4-NBA ERR is 38 mV dec$^{-1}$, far less than that for HER (90 mV dec$^{-1}$) (Supplementary Fig. 10a), indicating the more favorable kinetics of the critical role of FeP-MoP heterojunction in promoting electrocatalytic performance. A series of tests demonstrate that FeP-MoP/FF synthesized with 2 mM APM

manifests an optimized activity toward 4-NBA ERR (Supplementary Fig. 11). Electrochemical surface area (ESCA) was compared by measuring the double-layer capacitance (C$_{dl}$) via cyclic voltammetry (CV) (Supplementary Fig. 12). The C$_{dl}$ of FeP-MoP/FF is 151.2 mF cm$^{-2}$, larger than that of FeP/FF (74.5 mF cm$^{-2}$) and MoP/FF (38.4 mF cm$^{-2}$) (Fig. 3c), illustrating more exposed active sites of FeP-MoP/FF. Moreover, the activity normalized by ECSA for FeP-MoP/FF is also much higher than that for FeP/FF and MoP/FF (Supplementary Fig. 13), further suggesting the excellent intrinsic 4-NBA ERR activity of FeP-MoP heterojunction. Besides, electrochemical impedance spectroscopy (EIS) was performed. MoP-FeP/FF delivers a smaller charge transfer resistance (R$_{ct}$) of ~4.7 Ω than FeP/FF (~9.1 Ω) and MoP/FF (~13.9 Ω) (Fig. 3d), which implies the rapid charge transfer kinetics of FeP-MoP/FF.

The electroreduction of 4-NBA was carried out at the potential of 0.004 V vs. RHE, and 4-NBA substrate and its reduction products were examined by high performance liquid chromatography (HPLC). Nearly 116 C of charges is required to completely convert 4-NBA into ABA in 115 min (Fig. 3e). Figure 3f shows a decrease in 4-NBA concentration and an increase in 4-ABA concentration over time, further confirming the gradual conversion of 4-NBA to 4-ABA. Meanwhile, the by-products of 4-nitrosobenzyl alcohol (4-NSBA) and 4-hydroxyaminobenzyl alcohol (4-HABA) are negligible (Fig. 3g). FeP-MoP/FF shows an excellent 4-NBA ERR activity with high conversion, selectivity and faradaic efficiency (FE) of 99.5%, 99.1% and 99.0%, which outperforms FeP/FF, MoP/FF counterparts and recently reported catalysts (Fig. 3h and Supplementary Tables 1, 2). Stability is an important evaluating indicator for a catalyst. Impressively, FeP-MoP/FF maintains almost unchanged conversion, selectivity and FE for ten successive cycles (Fig. 3i). Furthermore, the morphology and structure of FeP-MoP/FF after 4-NBA ERR

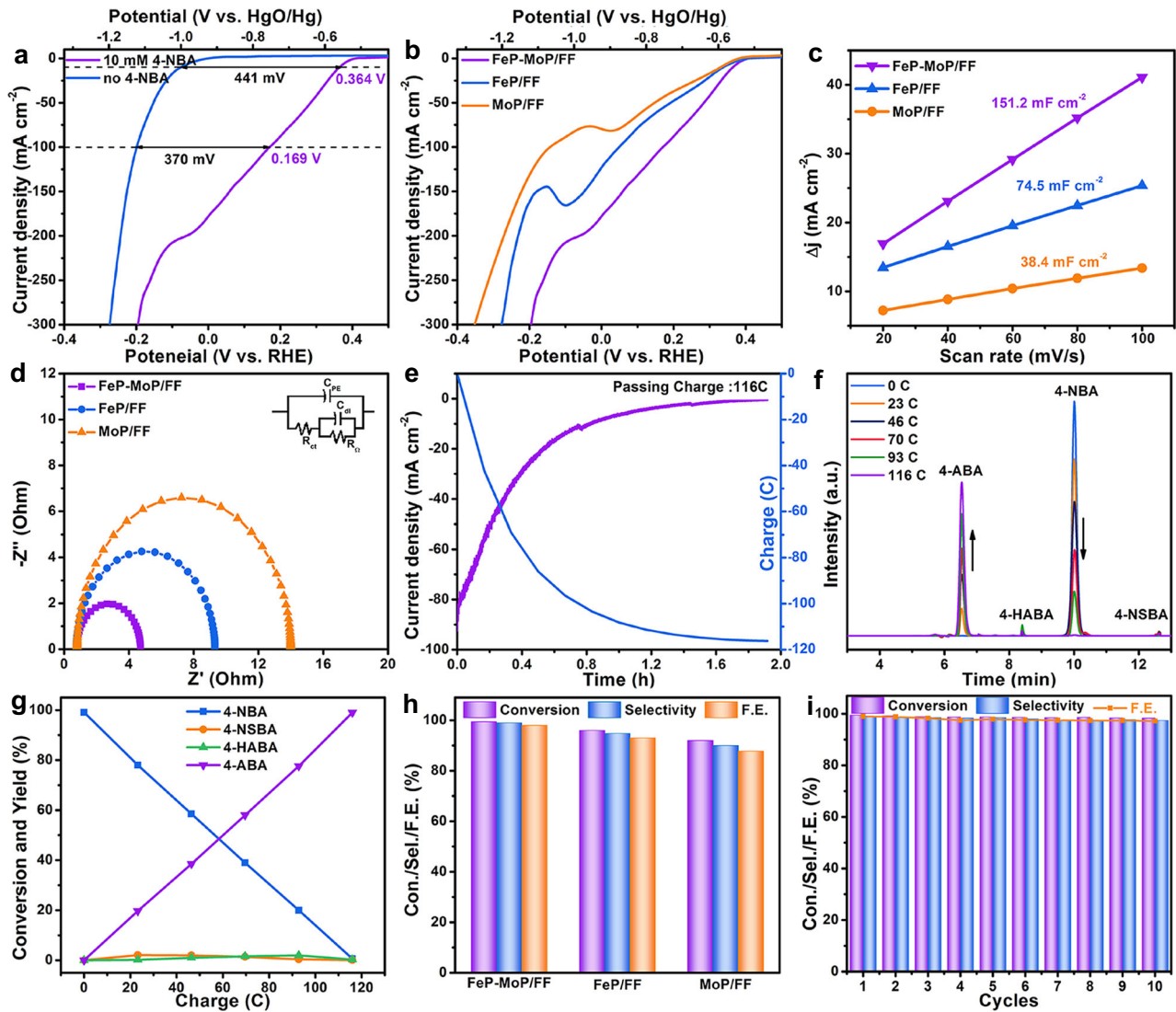

**Fig. 3 4-NBA ERR performance. a** LSV curves of FeP-MoP/FF in 1.0 M KOH without and with 10 mM 4-NBA at a scan rate of 5 mV s$^{-1}$. **b** LSV curves, **c** Capacitive curves, and **d** Nyquist plots of FeP-MoP/FF, FeP/FF and MoP/FF. **e** Chronoamperometric response curve. **f** HPLC-trace acquired at various charges. **g** Conversion and yield of reduction products over passed charges. **h** The conversion, selectivity and FE of 4-NBA ERR for FeP-MoP/FF, FeP/FF and MoP/FF **i** Consecutive use of FeP-MoP/FF for ten successive cycles.

show no apparent changes (Supplementary Fig. 14), suggesting the outstanding cycle stability of FeP-MoP/FF resulted from the structure rigidity. To further investigate the practical applications, a series of control experiments concerning the influence of reactant concentration, pH value and organic category on activity were performed (Supplementary Figs. 15–17 and Table 3), manifesting the good universality of FeP-MoP/FF for catalyzing electroreduction of various nitro compounds under different conditions.

**The 5-HMF EOR performance.** The 5-HMF EOR performance of FeP-NiMoP$_2$/FNF was evaluated in 1 M KOH with 10 mM 5-HMF. No the degradation of 5-HMF or Cannizzaro reaction occurs in the 1 M KOH solution (Supplementary Fig. 18). Figure 4a shows the LSV curves of FeP-NiMoP$_2$/FNF toward 5-HMF EOR (with 5-HMF) and OER (without 5-HMF). The potentials ($\eta_{10} = 1.333$ V, $\eta_{100} = 1.366$ V vs. RHE) for 5-HMF EOR, are 132 and 142 mV lower than those for OER. Additionally, the Tafel slope for 5-HMF EOR is 33 mV dec$^{-1}$, much smaller than that for OER (87 mV dec$^{-1}$) (Supplementary

Fig. 19a), suggesting that 5-HMF EOR is thermodynamically more favorable than OER. Besides, FeP-NiMoP$_2$/FNF outperforms FeP/FF and NiMoP$_2$/NF in performance (Fig. 4b, Supplementary Fig. 19b and Table 4), indicating the rapid 5-HMF EOR kinetics of FeP-NiMoP$_2$ heterojunction. Notably, FeP-NiMoP$_2$/FNF presents better performance than FeP-MoP/FF, which suggests the promoted role of Ni for boosting the 5-HMF EOR activity. Additionally, FeP-NiMoP$_2$/FNF prepared by adding 4 mM APM shows the best activity toward 5-HMF EOR (Supplementary Fig. 20). FeP-NiMoP$_2$/FNF shows a larger $C_{dl}$ (57.0 mF cm$^{-2}$) than NiMoP$_2$/NF (37.2 mF cm$^{-2}$), FeP-MoP/FF (30.1 mF cm$^{-2}$) and FeP/FF (16.5 mF cm$^{-2}$) (Fig. 4c and Supplementary Fig. 21), signifying that the heterostructure and nickel doping concurrently increase the active sites. FeP-NiMoP$_2$/FNF also delivers higher ECSA-normalized activity compared to the FeP-MoP/FF, FeP/FF and NiMoP$_2$/NF, further suggesting the superb intrinsic 5-HMF EOR activity of FeP-NiMoP$_2$/FNF heterojunction (Supplementary Fig. 22). FeP-NiMoP$_2$/FNF possesses the smallest $R_{ct}$ of ~5.0 Ω among FeP-MoP/FF (~12.4 Ω), FeP/FF (~20.1 Ω) and NiMoP$_2$/NF (~8.2 Ω) (Fig. 4d), suggesting the enhanced kinetics of FeP-NiMoP$_2$/FNF toward 5-HMF EOR.

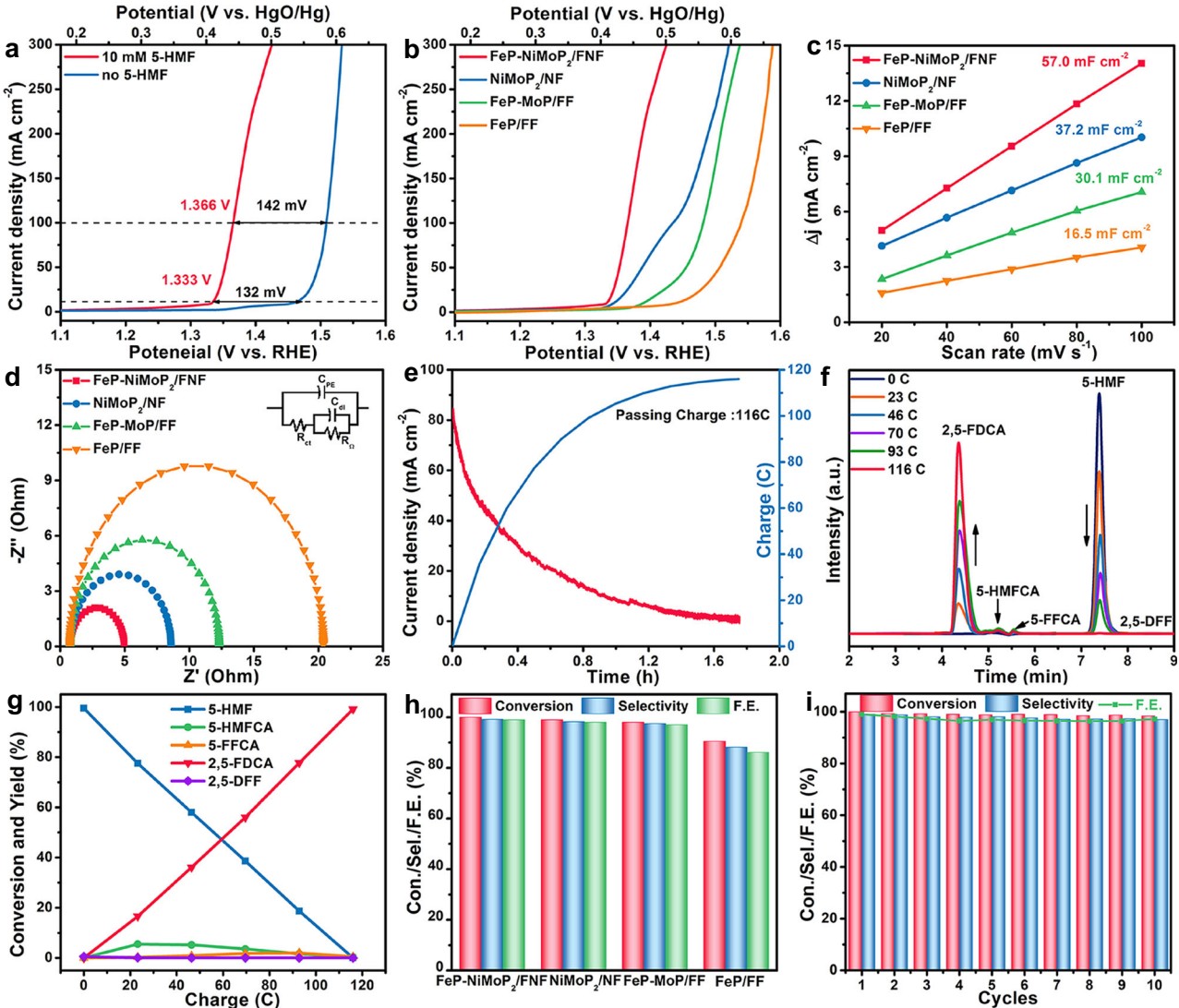

**Fig. 4 5-HMF ROR performance. a** LSV curves of FeP-NiMoP₂/FNF in 1.0 M KOH without and with 10 mM 5-HMF at a scan rate of 5 mV s⁻¹. **b** LSV curves, **c** Capacitive curves, and **d** Nyquist plots of FeP-NiMoP₂/FNF, NiMoP₂/NF, FeP-MoP/FF and FeP/FF. **e** Chronoamperometric response curve. **f** HPLC-trace acquired at various charges. **g** Conversion and yield of reduction products over passed charges. **h** The conversion, selectivity and FE of 5-HMF EOR for FeP-NiMoP₂/FNF, NiMoP₂/NF, FeP-MoP/FF and FeP/FF **i** Consecutive use of FeP-NiMoP₂/FNF for ten successive cycles.

The 5-HMF electrooxidation was carried out at 1.404 V vs. RHE, in which 5-HMF substrate and its oxidation products were analyzed. The theoretical charge is 116 C for complete transformation of 5-HMF to 2,5-FDCA (Fig. 4e). The charge is same with that of 4-NBA electroreduction, implying that 5-HMF EOR could be well coupled with 4-NBA ERR. The 5-HMF concentration gradually decreases and the 2,5-FDCA concentration increases over time (Fig. 4f). There are negligible by-products (2,5-diformylfuran (DFF), 5-hydroxymethyl furancarboxylic acid (5-HMFCA), and 5-formylfuran carboxylic acid (5-FFCA)) (Fig. 4g). Excitingly, FeP-NiMoP₂/FNF displays excellent 5-HMF EOR performance with high conversion, selectivity and FE of 100%, 99.2%, and 99.0%, far exceeding FeP-MoP/FF, FeP/FF and NiMoP₂/NF counterparts and most reported catalysts (Fig. 4h and Supplementary Tables 4, 5). Besides, FeP-NiMoP₂/FNF exhibits exceptional universality for catalyzing electrooxidation of various organics under different experimental conditions (Supplementary Figs. 23–25 and Table 6). Furthermore, FeP-NiMoP₂/FNF holds the excellent cycle stability (Fig. 4i). SEM and XRD analyses for FeP-NiMoP₂/FNF after 5-HMF EOR electrolysis demonstrate that this catalyst is still

FeP-NiMoP₂/FNF in nature with the preservation of its nanowire feature (Supplementary Fig. 26a, b). Further XPS analysis was performed. Metal elements (Fe, Mo, Ni) show no obvious change, but the P signal after 5-HMF EOR appears slightly weak (Supplementary Fig. 26c–f and Table 7), in which the peaks intensity related to Co-P bonds decrease, accompanied by the strengthened peak intensity assigned to the P-O bond. This validates that a certain surface reconstruction of FeP-NiMoP₂/FNF occurs and surface P is replaced by O to form the corresponding oxide during 5-HMF EOR and part of P may be leached. To probe the possibility of P leaching during electrocatalysis, the solution after 5-HMF EOR was analyzed by performing inductively coupled plasma optical emission spectrometer (ICP-OES). the content of metals are negligible, whereas P is detected with content of 5.6 μg mL⁻¹ (Supplementary Table 7), suggesting that a small amount of P leaches into solution, further ascertaining the replacement of P by O to form the corresponding oxide during 5-HMF EOR. The synergy of the phosphides and oxides formed during the 5-HMF EOR process can be cocurrently responsible for good 5-HMF EOR activity, which is in accordance with previous reports[14].

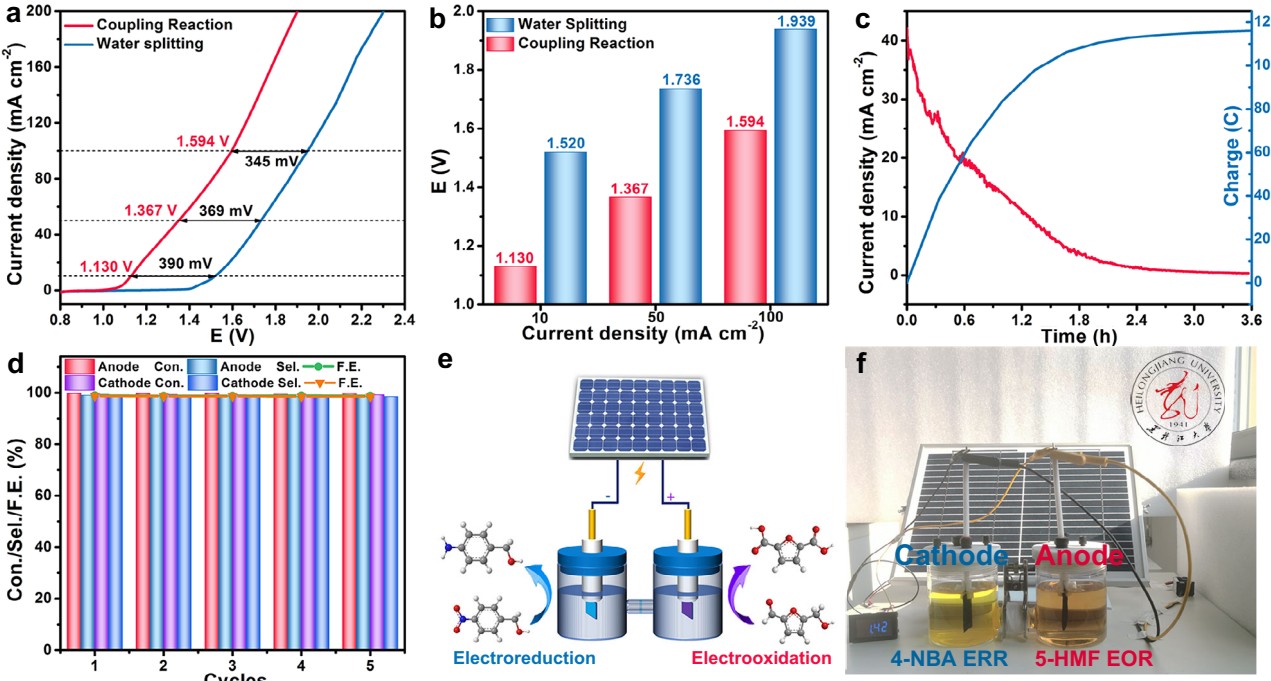

**Fig. 5 Paired electrocatalysis of 4-NBA ERR and 5-HMF EOR. a** LSV curves and **b** comparison of potentials of FeP-MoP/FF||FeP-NiMoP$_2$/FNF couple with and without organic substances. **c** Chronoamperometric response curve. at 1.404 V. **d** The conversion, selectivity and FE of anode and cathode during the coupling reaction for five successful cycles. **e** Illustration of the paired electrocatalysis system driven by solar-cell. **f** Demonstration of pairing 4-NBA ERR (left) and 5-HMF EOR (right) in H-type cell by using a solar-driven system with a voltage of 1.420 V.

**Paired electrolysis of 4-NBA ERR and 5-HMF EOR.** Assembling two desirable half-reactions into a paired electrolysis with efficient catalysts can maximize energy efficiency. 4-NBA ERR and 5-HMF EOR can be well coupled because the matchable reaction kinetics (Supplementary Fig. 27) and the equal number of transferred electrons guarantee synchronous 4-NBA reduction and 5-HMF oxidization, generating two target products. Furthermore, the only difference between FeP-MoP/FF and FeP-NiMoP$_2$/FNF is the Ni in FeP-NiMoP$_2$/FNF, which means little mutual interference during paired eletrocatalysis. Accordingly, the electrolyzer integrated by FeP-MoP/FF cathode and FeP-NiMoP$_2$/FNF anode for paired electrolysis of 4-NBA ERR and 5-HMF EOR system shows low voltages of 1.130, 1.367, and 1.594 V at 10, 50, and 100 mA cm$^{-2}$ (Fig. 5a, b), 390, 369, and 345 mV lower than those for water splitting, indicating the higher efficiency of the paired electrolysis system than water splitting. When applied voltage of 1.400 V, the 4-NBA and 5-HMF substrates can be converted rapidly and completely within 215 min (Fig. 5c), implying the rapid reaction kinetics. The paired electrolysis system also shows high conversion, selectivity and FE of ≥99%, as well as good cycle stability for both sides (Fig. 5d and Supplementary Fig. 28). Impressively, the electrolyzer can be driven by a solar cell with a voltage of 1.420 V to accomplish the simultaneous 4-NBA ERR and 5-HMF EOR (Fig. 5e, f), which manifests its huge potential to couple with solar energy.

## Discussion

**Study on the catalytic mechanism.** To disclose the active origins and identify the catalytic mechanisms of Fe-Mo-based phosphides toward 4-NBA ERR and 5-HMF EOR, density functional theory (DFT) calculation was performed. The optimized models of FeP, MoP, NiMoP$_2$, FeP-MoP, and FeP-NiMoP$_2$ systems were established. The FeP (211) and MoP (100) and NiMoP$_2$ (001) crystal planes were applied to build the theoretical models based on their exposed planes in TEM, where FeP and MoP (or

NiMoP$_2$) can be well matched when their ratio is commensurated 3:2 (Supplementary Fig. 29).

*4-NBA ERR catalytic mechanism.* FeP-MoP holds a higher density of state (DOS) near the Fermi level (E$_F$) than FeP and MoP, which facilitates rapid electron transfer in electrocatalysis (Supplementary Fig. 30a). In addition, surface work function (W$_F$) of catalysts was obtained by calculating the vacuum level and Fermi level. (Supplementary Fig. 30b). The lowest W$_F$ of FeP-MoP indicates the easiest to reduce 4-NBA on FeP-MoP surface than other counterparts[45]. The adsorption energies of 4-NBA (E$_{4-NBA}$) on FeP-MoP, FeP and MoP are 1.33, 1.15, and 0.82 eV, respectively (Fig. 6a and Supplementary Table 8). The highest E$_{4-NBA}$ suggests FeP-MoP heterojunction owns most favorable kinetics for 4-NBA ERR. Fe site possesses the largest E$_{4-NBA}$ among various sites in FeP-MoP heterojunction. This result could be explained by calculating the different charge density of FeP-MoP heterojunction. As depicted in Fig. 6b, the electrons (1.76 e$^-$) transfer from MoP to FeP at FeP-MoP interface, resulting in electron accumulation on Fe site and electron depletion on Mo site, which agrees with the XPS analysis. Electron accumulation on Fe site can provide more electrons for reduction of 4-NBA. Thus the Fe site in FeP-MoP heterojunction is identified as the active center for 4-NBA ERR. Additionally, the E$_{4-NBA}$ of FeP-NiMoP$_2$ was calculated as (−1.29 eV), which is smaller than that of FeP-MoP, indicating the unfavorable 4-NBA ERR kinetics of introducing Ni (Supplementary Table 8) and is accordance with experimental results. It can be ascribed to that when introducing Ni, it traps electrons from Fe, resulting into the less electrons on Fe site, which is not conductive for the electroreduction of 4-NBA. Furthermore, the reaction mechanism of FeP-MoP heterojuction for catalyzing 4-NBA ERR is deduced (Fig. 6c). 4-NBA ERR undergoes three two-electron-proton processes (Supplementary Fig. 31). 4-NBA is adsorbed on Fe site and 4-NBA* is reduced to 4-NBA + H*. Afterward, two different situations with

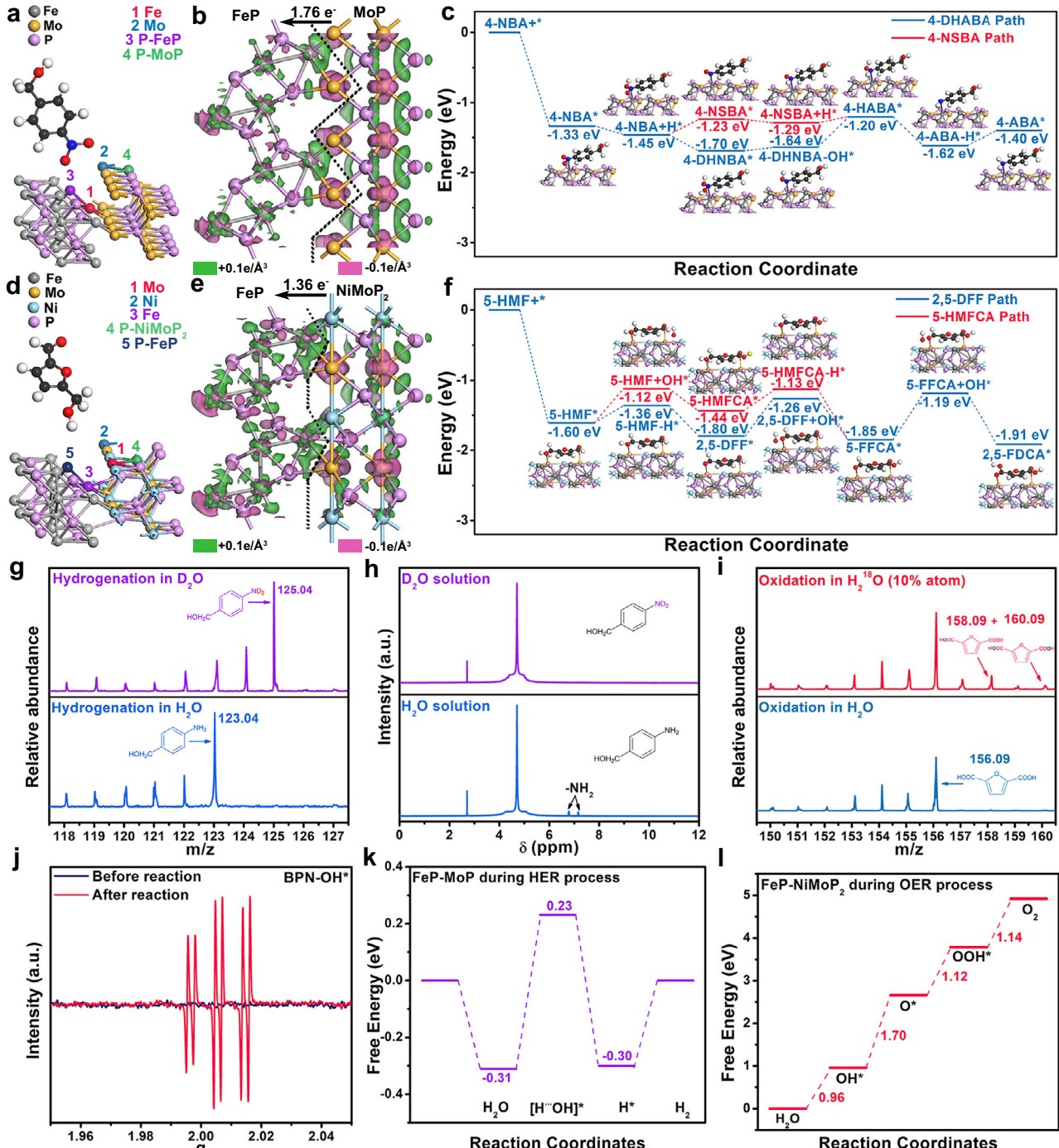

**Fig. 6 DFT theoretical calculation, and study of water as sole H and O sources. a** Possible adsorption sites of 4-NBA on the surface of FeP-MoP. **b** Electron density difference plot of FeP-MoP, **c** Energy profiles of 4-NBA reduction to 4-ABA on the FeP-MoP surface. **d** The possible adsorption sites of 5-HMF on the surface of FeP-NiMoP₂. **e** Electron density difference plot of FeP-NiMoP₂. **f** Energy profiles of 5-HMF oxidation to 2,5-FDCA on the FeP-NiMoP₂ surface. **g** ESM and **h** $^1$H-NMR spectra of the final product of 4-NBA EHR in H₂O (bottom) and D₂O (top) solution. **i** ESM spectra of the final product of 5-HMF EOR product in H₂O (bottom) and H₂¹⁸O (top) solution. **j** ESR detection of PBN-•OH for FeP-NiMoP₂/FNF in H₂O solution before and after 5-HMF EOR. **k** Free energy diagram for HER on the FeP-MoP. **l** Free energy diagram for OER on FeP-NiMoP₂.

the 4-NSBA* and 4-DHABA* intermediates are involved[46]. In the 4-NSBA path, 4-NBA + H* is reduced to 4-NSBA*, 4-NSBA + H*, 4-HANA*, 4-ABA-H*, and 4-ABA* in sequence. The difference of 4-DHNBA path from 4-NSBA path is that 4-NBA + H* is reduced to 4-DHABA* and 4-DHABA-OH* in sequence. In view of thermodynamics, 4-NSBA* intermediate is easier to desorb, and the activation energy of rate-determining

step (RDS) in 4-NSBA path (4-NBA + H* → 4-NSBA*, 0.22 eV) is much smaller than that of 4-DHABA path (4-DHABA-OH* → 4-HABA*, 0.44 eV) (Fig. 6c). This suggest that 4-NBA ERR is more inclined to adopt 4-NSBA path. Furthermore, electrospray mass spectrometry (EMS) analysis further confirms that FeP-MoP heterojuction undergoes the 4-NSBA path (Supplementary Fig. 32).

*5-HMF EOR catalytic mechanism.* The FeP-NiMoP$_2$ heterojunction shows larger DOS than FeP-MoP, FeP and NiMoP$_2$ at the E$_F$, demonstrating the rapid electron transfer of FeP-NiMoP$_2$ heterojunction (Supplementary Fig. 33a). FeP-NiMoP$_2$ has the lowest W$_F$ for 5-HMF EOR (Supplementary Fig. 33b), indicating 5-HMF oxidation on FeP-NiMoP$_2$ surface is the easiest compared with other samples[47]. Accordingly, FeP-NiMoP$_2$ (Mo site) shows the highest 5-HMF adsorption energy (E$_{5-HMF}$) compared with FeP-MoP, NiMoP$_2$ and FeP (Fig. 6d and Supplementary Table 9), demonstrating most favorable kinetics for 5-HMF EOR on FeP-NiMoP$_2$. Based on charge analysis, the electrons of Mo transfer to Fe (1.36 e$^-$) and Ni (0.88 e$^-$), respectively. The total electron loss of Mo is 2.24 e$^-$ in FeP-NiMoP$_2$, more than that in FeP-MoP (1.76 e$^-$). The more electron-deficient Mo benefits the adsorption of 5-HMF, and thus Mo site is the active center in FeP-NiMoP$_2$ heterojunction for 5-HMF EOR. The catalytic mechanism of 5-HMF EOR over FeP-NiMoP$_2$ (Mo site) is also induced (Fig. 6e). The oxidation of 5-HMF to 2,5-FDCA has two pathways, including aldehyde oxidation to give 5-HMFCA and hydroxyl methyl oxidation to generate 2,5-DFF (Supplementary Fig. 34)[28]. In the 5-HMFCA path, 5-HMF can be adsorbed on Mo site where 5-HMF* is oxidized to 5-HMFCA* by OH* free radical and then further oxidized to 5-FFCA* and 2,5-FDCA in sequence. The difference of 2,5-DFF path from 5-HMFCA path is that 5-HMF* is oxidized to 2,5-DFF*. The activation energy of RDS (5-HMF* → 5-HMFCA*, 0.48 eV) in 5-HMFCA path is lower than that in 2,5-DFF path (2,5-DFF* → 5-FFCA*, 0.54 eV) (Fig. 6f), suggests that the 5-HMF oxidation follows the 5-HMFCA path, which is further confirmed by EMS results (Supplementary Fig. 35). Furthermore, to elucidate the role of reconstructed oxide formed during 5-HMF EOR electrocatalysis on performance, O-substituted FeP-NiMoP$_2$ model (FeP-NiMoP$_2$-oxi) was established by replacing a part of P by O. As shown, FeP-NiMoP$_2$-oxi possesses the lower W$_F$ (3.50 eV) and higher E$_{5-HMF}$ (−1.63 eV) compared to FeP-NiMoP$_2$, further indicating the important role of O species in the catalyst on promoting the 5-HMF EOR (Supplementary Fig. 36 and Table 9).

*Study on water as sole hydrogen and oxygen sources.* To elucidate the sources of hydrogen and oxygen in eletroreduction and eletrooxidation reactions, the isotope labeling technology was performed[48,49]. When D$_2$O substitutes for H$_2$O to mix with 4-NBA, D atoms in the final product 4-ABA are detected, suggesting that the hydrogen comes from water in the 4-NBA ERR (Fig. 6g). Additionally, the signals of $^1$H-NMR at chemical shift of 6.8 and 7.1 ppm corresponding to amino group (-NH$_2$) disappear after replacing H$_2$O by D$_2$O (Fig. 6h)[50], further confirming the source of hydrogen is water. Likewise, when O atoms in the solution is marked by H$_2$$^{18}$O, $^{18}$O atoms in final product 2,5-FDCA are detected, which suggests the source of oxygen also comes from water in the 5-HMF EOR (Fig. 6i). Considering the existence of various oxygen species (OH*, OOH*, and O*), electron spin resonance (ESR) spectrum was carried out to verify which oxygen species participate in the oxidation reaction. As displayed in Fig. 6j, FeP-NiMoP$_2$/FNF displays a set of intense peaks after 5-HMF EOR that matches perfectly with PBN-•OH radicals, illustrating that OH* free radicals are active species for 5-HMF EOR[51]. Water as sole H and O source for the paired system is a sustainable and recyclable route. Because the valuable target products could be continuously and efficiently produced by the addition of the reactants at two-electrode chambers without producing side products. Otherwise, if H or O species come from organics, such as H provided by hydroxyl groups of orgaincs[52] or dehydrogenation of anodic oxidation[53] and O originated from deoxygenation of cathodic reduction[54], other reaction pathways occur and side reactions happen, resulting in reducing the

selectivity of target products[55]. Or if dissolved oxygen from air as O source is also not sustainable due to the small amount[56].

*Intrinsic superiority of the paired electrocatalysis of organics.* To further elaborate the intrinsic advantages of the paired electrocatalysis of organics over water splitting, we also investigate the water splitting process. The HER in alkaline media can be described by three steps, including water dissociation to generate H* intermediates, desorption of *OH intermediates, and formation of hydrogen (Fig. 6k)[57]. In the 4-NBA ERR, only first step that water dissociation generates H* in HER is involved without requiring extra energy for subsequent H$_2$ production. Moreover, the energy barrier of the whole reaction for 4-NBA ERR is 0.22 eV, much lower than that of HER (0.52 eV)[58]. Hence, 4-NBA ERR has more favorable kinetics than HER. OER involves four electron-transfer steps (Fig. 6l)[59]. The first step (H$_2$O to OH*) is required for 5-HMF EOR. Particularly, the ΔG$_{H2O-OH}$ shows the smallest energy difference (0.96 eV) compared with the subsequent three steps. Therefore, the electrooxidation of 5-HMF by OH* species effectively avoids the huge energy barriers for O$_2$ production. Additionally, the energy barrier of the whole reaction for 5-HMF EOR (0.66 eV) is far lower than that of OER (1.70 eV). The aforementioned results illustrate that 5-HMF oxidation is more competitive than OER[60]. Based on the above analysis, the integration of 4-NBA ERR with 5-HMF EOR substantially decreases energy consumption in comparison with water splitting.

In summary, we have designed the FeP-MoP/FF and FeP-NiMoP$_2$/FNF electrodes for effectively paired electroreduction and electrooxidation of organics with water as the sole feedstock. The catalytic mechanisms and intrinsic superiority of the paired electrocatalysis of organics on Fe-Mo-based phosphide heterojunctions are clearly elaborated by a combination of the experimental and theoretical methods. The enhanced electronic conductivity and electron transfer of FeP-MoP and FeP-NiMoP$_2$ heterojunctions optimize the of 4-NBA ERR and 5-HMF EOR reaction kinetics, contributing to the superior 4-NBA ERR and 5-HMF EOR activities. The paired electrolyzer coupled with solar cell exhibits low electrolytic voltages, high faradaic efficiency and good cycle stability. This work opens a new avenue for sustainable synthesis of various value-added organic chemicals with cheap water as a feedstock.

## Methods

**Chemicals.** Potassium carbate (K$_2$CO$_3$), potassium hydroxide (KOH), acetone, hydrochloric acid (HCl) and sodium hypoposphite (NaH$_2$PO$_2$) were purchased from Kermel chemical reagent Co. Ltd. Ammonium heptamolybdate (APM) was from Sinopharm reagent Co. Ltd. Ascorbic acid (C$_6$H$_8$O$_6$), 2-furoic acid (FA), 5-hydroxymethylfurfural (5-HMF), 2,5-furandicarboxylic acid (2,5-FDCA), 2,5-diformylfuran (2,5-DFF), 5-formylfurancarboxylic acid (5-FFCA), 5-hydroxymethyl furancarboxylic acid (5-HMFCA), benzyl alcohol (BA), benzaldehyde (BH), benzoic acid (BZA), furfural (FF), furfuryl alcohol (FFA), 4-nitrobenzyl alcohol (4-NBA), 4-aminebenzyl alcohol (4-ABA), 4-nitrobenzoic acid (4-NBZA), 4-aminebenzyl alcohol (4-MBA), 4-methoxybenzyldehyde (4-MBH), 4-methoxybenzoic acid (4-MBZA), 3-nitrobenzyl alcohol (3-NBA) and 2-nitrobenzyl alcohol (2-NBA) were purchased from Aladdin Chemical Co. Ltd. Iron foam (FF), iron nickel foam (FNF) and nickel foam (NF) were purchased from Kunshan Jiayisheng Electronics Co. Ltd. Deionized (DI) water was used in all experiments.

**Synthesis of FeP-MoP/FF.** 2 mM (NH$_4$)$_6$Mo$_7$O$_{24}$·4H$_2$O (APM), 14 mM ascorbic acid, and 50 mL mixture solution of deionized water and ethylene glycol (4:1) were ultrasonically mixed for 10 min. After stirring for 1 h, the mixture was transformed into a 100 mL Teflon-lined autoclave, and one piece of FF (3*4 cm$^2$) was added in this solution and hydrothermally treated at 160 °C for 10 h. The resultant FF slice (brown-black) was sonicated, washed repeatedly with deionized water and ethanol, and then put in an oven at 60 °C for 3 h. Then, the obtained FF slice was phosphatized at 600 °C for 1 h with a ramp rate of 5 °C min$^{-1}$ using NaH$_2$PO$_2$ as phosphorous source. For comparison, FeP/FF was synthesized by the similar synthetic procedure of FeP-MoP/FF without adding Mo source. MoP powder was

firstly synthesized and then coated on the FF substrate (denoted as MoP/FF). The FeP-MoP/FF samples with different Mo concentration were prepared by regulating the concentration of the APM from 1 to 5 mM.

**Synthesis of FeP-NiMoP₂/FNF**. 4 mM APM, 16 mM ascorbic acid, and 50 mL mixture solution of DI water and ethylene glycol (4:1) were ultrasonically mixed for 10 min. After stirring for 1 h, the mixture was transformed into a 100 mL Teflon-lined autoclave, and one piece of FNF ($3*4 \, cm^{-2}$) was added in this system and hydrothermally treated 150 ºC for 12 h. The resultant FNF slice (dark brown) was sonicated, washed repeatedly with deionized water and ethanol, and then kept in an oven at 60 ºC for 3 h. Afterward, the FNF slice $NaH_2PO_2$ was phosphatized at 500 °C for 1 h with a ramp rate of 5 °C $min^{-1}$ using $NaH_2PO_2$ as phosphorous source. For comparison, replacing FNF by Ni foam (NF), $NiMoP_2$/NF was synthesized by the similar synthetic procedure of FeP-NiMoP₂/FNF. The FeP-NiMoP₂/FNF samples with different Mo concentration were prepared by regulating the concentration of the APM from 1 to 5 mM.

**Electrochemical measurements**. The ERR and EOR measurements were performed with a three-electrode system in a H-type electrochemical cell by using a CHI electrochemical workstation (Model 760E). The synthesized materials were cut into 1 × 1 cm as the working electrodes. Hg/HgO electrode and carbon rod were used as reference electrode and counter electrode, respectively. All of the reported potentials were quoted with respect to the reversible hydrogen electrode (RHE) through $E_{RHE} = E_{HgO/Hg} + 0.059 \times pH + 0.098$ V. Cyclic voltammetry (CV) was conducted at a scan rate of 100 mV $s^{-1}$ to activate the electrodes. The LSV curves were obtained by scanning at a scan rate of 5 mV $s^{-1}$. All the collected data were corrected for 95% iR compensation. The 4-NBA ERR tests were conducted in 1.0 M KOH solution with 10 mM 4-NBA. The 5-HMF EOR were similar to ERR, except that the electrolyte was 1.0 M KOH solution with 10 mM 5-HMF. A paired electrolyzer was performed on an H-cell in which the two chambers were separated by an anion exchange membrane (Nafion 117). Electrochemically surface area (ECSA) was calculated from scan-rate dependence of CVs with scan rates of 10–100 mV $s^{-1}$. The electrochemical impedance spectroscopy (EIS) measurements were carried out with frequency from 0.01 Hz to 100 kHz at an amplitude of 10 mV. The other pH electrolytes were prepared by using the buffer solutions, in which the $K_2HPO_4$-$K_3PO_4$ buffer solution for pH = 11.63, $K_2CO_3$-$KHCO_3$ buffer solutions for pH = 10.02 and 8.31 and $K_2HPO_4$-$KH_2PO_4$ buffer solution for pH = 6.86. The activities of 4-NBA ERR and 5-HMF EOR in these pH electrolytes were measured.

**Product quantification**. To analyze the products of biomass quantitatively and calculate the corresponding Faradaic efficiency (FE), 20 μL electrolyte solution during chronoamperometry at 0.004 V (for ERR) or at 1.404 V (for EOR) was taken out from the electrolyte solution and diluted with 380 μL 0.1 M HCl solution, which was then analyzed using HPLC at 30 °C. The HPLC was equipped with a 4.6 × 250 mm Shim-pack GWS 5 μm C 18 column. The ultraviolet-visible detector was set at 254 nm for the electroreduction of 4-NBA, 3-NBA, 2-NBA, 4-NAS, 4-NBZA, and 4-NP. And the isocratic elution of 40% 0.02 M ammonium acerate solution and 60% methanol for electroreduction of 4-NBA, 3-NBA, 2-NBA, 4-NAS, 4-NBZA, and 4-NP for 14 min at the flow rate was set at 0.5 mL $min^{-1}$. A mixture of eluting solvent (A and B) was utilized. Solvent A was 5 mM ammonium formate aqueous solution and solvent B was methanol. Separation and quantification were accomplished using an isocratic elution of 70% A and 30% B for electrooxidation of HMF, FF, and FFA for 12 min and 40% A and 60% B for electrooxidation of BA, 4-NBA and 4-MBA for 15 min run time and both of the flow rate was set at 0.5 mL $min^{-1}$. For the product quantification of electrooxidation, the ultraviolet-visible detector was set at 265 nm for the electrooxidation of HMF, FF, and FFA and at 254 nm for the electrooxidation of BA, MBA, 4-NBA, and all the mentioned nitro aromatic compounds, respectively. Identification and quantification of the products were determined from the calibration curves by applying standard solution with known concentration of commercially purchased pure reactants, intermediates, and final products.

The conversion (%) and the selectivity (%) of the products were calculated using Eqs. (1) and (2):

$$Conversion = \frac{mol \, of \, organic \, consumed}{mol \, of \, initial \, organic} \times 100\% \quad (1)$$

$$Selectivity(\%) = \frac{mol \, of \, product \, formed}{mol \, of \, initial \, organic} \times 100\% \quad (2)$$

The Faradaic efficiency of product formation was calculated using the Eq. (3):

$$FE(\%) = \frac{mol \, of \, product \, formed}{total \, charged \, passed \div nF} \times 100\% \quad (3)$$

where F is the Faraday constant (96485 C $mol^{-1}$) and n is the electron transfer number.

**Theoretical calculation**. Density Functional Theory (DFT) simulations were performed by employing Cambridge Sequential Total Energy Package (CASTEP)

module implemented in Material Studio. The generalized gradient approximation (GGA) with a Perdew-Burke-Ernzerhof (PBE) functional was used to describe the electronic exchange and correlation effects. The 3 × 2 FeP supercell, 2 × 2 MoP and 2 × 2 NiMoP₂ were used as crystal models for facet cleavage and construction of heterojunctions based on TEM analysis.and a four-layer of slab supercell was chosen as the surface slab supercell to decrease the complexity of calculation. For FeP-MoP surfaces, 3 × 2 FeP (211) slab supercell (u = 13.44 Å v = 13.14 Å θ = 120°) and 2 × 2 MoP (100) slab supercell (u = 12.89 Å v = 12.89 Å θ = 120°) was applied to construct the FeP-MoP heterojunction model. Similarly, 3 × 2 FeP (211) slab supercell (u = 13.44 Å v = 13.14 Å θ = 120°) and 2 × 2 NiMoP₂ (001) slab supercell (u = 13.31 Å v = 13.31 Å θ = 120°) was applied to construct the FeP-NiMoP₂ heterojunction model. Besides, the model of FeP-NiMoP₂ with the P atoms on the surface replaced with O (denoted as FeP-NiMoP₂-oxi) was established. Typically, the four-coordinated P atoms forming tetrahedra on the surface of FeP phase and six-coordinated P atoms forming octahedra on the surface of NiMoP₂ phase were replaced to construct the O replaced model FeP-NiMoP₂-oxi. Population analysis was performed by the Mulliken scheme. and the plane-wave energy cutoff was set to 489.8 eV. Furthermore, the sampling over Brillouin zone was treated by a (2 × 2 × 1) Monkhorst-Pack grid, and a vacuum slab with the length of 15 Å was placed along z axis on each slab to avoid the pseudo interactions between periodic images. Geometry optimization was repeated until the total energy tolerance was converged to $2 \times 10^{-6}$ eV and the changes of the force on the atoms less than 0.05 eV/Å. The adsorption energy of organic substance was calculated by using the Eq. (4):

$$E_{ads} = E_{total} - (E_{surface} + E_{absorbate}) \quad (4)$$

where $E_{total}$, $E_{surface}$, and $E_{substrate}$ are the total energy of the adsorption state system, the total energy of the pure surface and the total energy of organic substance, respectively.

**Physical characterizations**. The scanning electron microscopy (SEM) test was conducted on a Hitachi S-4800 instrument at an accelerating voltage at 5KV. Transmission electron microscopy (TEM) characterization was carried out on a JEM-2100 electron microscope (JEOL, Japan) with an acceleration of 200 kV. X-ray diffraction (XRD) patterns were carried out using a Bruke D8 equipped with Cu-Ka radiation (1.5406 Å). X-ray photon spectroscopy (XPS) analysis was carried out using a VG ESCALABMK II with Mg-Ka radiation (1253.6 eV). The inductively coupled plasma optical emission spectrometer analysis (ICP-OES) was carried by a Agilent 720 ICP-OES With a RF transmitter at the power of 1.0 kW. The organics were quantified by high-performance liquid chromatography (HPLC, Wufeng LC-100C). The HPLC was equipped with an ultraviolet-visible detector set at 265 nm and a 4.6 × 250 mm Shim-pack GWS 5 μm C 18 column. $^1$H-NMR spectra were recorded using a Varian Mercury plus 400NB spectrometer relative to tetramethylsilane (TMS) as internal standard. Molecular masses were determined by a FINNIGAN LCQ Electro-Spraying Mass Spectrometry (EMS) The Electron spin resonance (ESR) test was determined by a Bruker EMXplus with the microwave power between 10 μW and 100 mW and a modulation amplitude of 1.00 G.

## Data availability
The data that support the plots within this paper are available of this study are available in the Source data file. The source data of Figs. 2a–f, 3a–i, 4a–i, 5a–f, 6c, f–l and Supplementary Figs. 26c–f are provided as a Source Data file. Other data is available from the corresponding author upon reasonable request. Source data are provided in this paper.

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

## Acknowledgements

This research was supported by the National Key R&D Program of China (2018YFB1502401, H.F.), the National Natural Science Foundation of China (U20A20250, H.F., 22171074, H.Y., 21631004, H.F., 21805073, H.Y., 91961111, C.T., 21901064, A.W.), the Heilongjiang Provincial Natural Science Foundation of China (YQ2021B009, H.Y.), and the Basic Research Fund of Heilongjiang University in Heilongjiang Province (2021-KYYWF-0031, H.Y.).

## Author contributions

H.Y. and H.F. conceived the idea. G.Y. performed the experiments. A.W. and C.T. performed the SEM and TEM characterizations. G.Y. and Y.X. performed the DFT calculations studies. Y.W. helped with electrochemistry experiments. H.Y. and Y.J. designed and revised the structure and logic of the manuscript. G.Y., H.Y., and H.F. wrote the manuscript with input from all co-authors.

## Competing interests

The authors declare no competing interests.
