## [Peer review file · Nature Communications]

REVIEWER COMMENTS

Reviewer #1 (Remarks to the Author):

In the paper, Yang et al. reported Fe-Mo-based phosphide heterojunctions for both 4-NBA reduction and HMF oxidation. The catalysts were well-characterized and reaction pathways were well-investigated. However, these reactions on non-noble catalysts and their reaction pathways have been well-reported in the literature, such as, ACS Sustainable Chemistry & Engineering 9.5 (2021): 1970-1993, CCS Chem. 7, 507-515 (2020), Angew. Chem. Int. Ed. 58, 9155-9159 (2019). Considering the novelty of this work, it is hard to be accepted by Nature Communications.

Other comments are shown below.

1. Line 11, the authors mentioned “active H* species for organic electroreduction originate from water.” Why did they want to highlight this in the abstract? Is there any other possible H source in the aqueous solution?
2. Line 28-29, “both hydrogen and oxygen evolution reactions (HER and OER) are slow kinetics”. We usually agree on that only OER is kinetically sluggish, which is major barrier to water electrolysis.
3. Line 102-103, how did the authors test the nanosheet thickness by TEM? Why not just using the cross-section SEM to measure the thickness?
4. Line 136-137, should it be called magnetic FeNi₃ or alloy FeNi₃? It is not the metallic, right?
5. Line 161-162, “Such electron-rich Fe is similar to that of nitroreductase, 28 which is conducive to 4-NBA ERR.” The citation 28 is on nitrogen reduction to ammonia, not 4-NBA ERR. So what is the similarity of these two works and how transferable and comparable of the similar catalysts activity?
6. Line 174-176, “where FeP-MoP/FF exhibits a better 4-NBA ERR activity than FeP-NiMoP₂/FNF (Supplementary Fig. 9).” What could be the reason to explain FeP-MoP/FF exhibits higher activity? Is it because of the higher surface area or any other possibilities? Did the authors test electrochemical active surface area (ECSA)?

7. Figure 3b-c, Figure 3h, and line 197-199. The authors tested the LSV and ECSA among three catalysts. It seems that the higher activity of FeP-MoP/FF than FeP/FF and MoP/FF is mainly originated from the higher surface area of FeP-MoP/FF? What is the ECSA-normalized reaction activity (or current density)?

8. Line 203, "the electroreduction of 4-NBA was carried out at the potential of 0.004 V". Is it 0.004 V vs. RHE? Why did the authors select so positive potential for electroreduction? Please check this value.

9. Similar questions as #7 for FeP-NiMoP2/FNF to catalyze 5-HMF EOR.

10. Line 352-353, DFT results showed that "Based on charge analysis, the electrons of Mo transfer to Fe (1.36 e-) and Ni (0.88 e-), respectively." This conflicts with the XPS results (line 167), Ni captures electrons from Fe and Mo. How to explain this inconsistency?

11. Line 367-379, the authors used isotopic labelling and confirmed the proton and oxygen source are both from water. Are there any other possibilities of proton/oxygen sources? Based on previous literatures and the high conversion and selectivity in this work, the H/O source would not be coming from the contaminations, it is obviously from H₂O to provide H/O source.

12. HMF is not stable in strong alkaline solutions, e.g., 1 M KOH, ACS Catal. 2018, 8, 2, 1197–1206. Did the authors observe the degradation of HMF or Cannizzaro reaction in the electrolyte?

13. Figure S15 and S22. How did the authors prepare different pH electrolytes? Did the authors use buffer solutions? If not, the electrolyte would quickly change to alkaline / acidic solution during ERR (proton consumption reaction) and EOR (proton generation reaction) reactions, respectively. So that the pH of electrolytes before and after reactions would show large differences. Therefore, the calculation of potential versus RHE and explanation to reaction mechanisms may have problems.

Reviewer #2 (Remarks to the Author):

The authors have designed a paired electrolysis system using Fe-Mo based electrodes for simultaneously reduction and oxidation of organics with water as the feedstocks, which can be driven by a solar cell with extremely low electrolytic voltages and high Faradaic efficiency. This work will be highly interested

for the researchers in electrocatalysis, electrosynthesis, green chemistry, and energy conversion. In addition, this manuscript is well organized and well presented. Thus, this interesting manuscript deserves being published in Nature Communications after addressing the following minor concerns.

1. The sentence “Ni based material serve as the catalysts for 5-HMF EOR” on line 68-69 stands out here, which is logically incoherent with the context. The authors is encouraged to rewrite this part to expand the elaboration.
2. In Fig. 3b, the yellow line emerges a sharp current increase at about 0.3 V, making the turning point presents a strange, uneven trend like an obtuse angle. Similarly, it is also observed on the green line in supplementary Fig. 11. It’s rarely seen in the LSV test. It means some oxidation peaks? Please explain them briefly or provide new data.
3. The statement “Only Sun’s group ...” on line 58 is inaccurate. Actually, paired electrolysis strategy has been applied in some early works (e.g. Ref. 25; doi: 10.1002/anie.202009155; doi: 10.1002/anie.202009757). Thus, it is suggested to revise the related description. The author list in Ref. 25 is wrong. Please double check the author lists of Ref.
4. All the potentials normalized to RHE may be inaccurate because the pH value often vary during the water-involving organic reaction. Generally, all the electrochemical data vs the actual reference electrode should be provided in the text (see the ref. Nature Communications 2021, 12, 3881). So, the data vs Hg/HgO are suggested in the manuscript, or the potential vs Hg/HgO electrode should be add to Fig. 3a, the fist electrochemical figure.
5. In Fig. 5a, how is the voltage of the two-electrode electrolytic system converted into the relative RHE potential? Here I think it is the cell voltage. "Potential (V vs. RHE) " should be correct to be "Cell voltage (V vs. counter electrode)", "E (V vs. counter electrode)" or "E (V)".

Reviewer #3 (Remarks to the Author):

Yang and coworkers presented a family of electrocatalysts that reduce/oxidize organic molecules with very high faradaic efficiencies, compared to the background hydrogen and oxygen evolution reactions (HER/OER). FeP-MoP supported on Fe foam (FF) and a similar material containing Ni (FeP-NiMoP2/FNF) were the best performers for reduction of 4-NBA and oxidation of 5-HMF respectively. The combined oxidation/reduction of such organic molecules demonstrate an appealing path to be further explored. Before publication, however, there are blind spots that need clarification.

1. The main one I currently observe is related to the models in Suppl Fig 26, specifically FeP-NiMoP2 (e). It seems that the authors produced it from that of FeP-MoP (d). The specific details about how the authors produced both of them should be dully explained. For instance, it seems that they

commensurated FeP and MoP in a 3:2 ratio (Figure 1d), but that is not said anywhere (eg, lines 107-108 and 120 do not say that). Same with NiMoP₂ (Figure 1h).

2. Can these surfaces reconstruct upon cleavage? NiMoP₂(001) and MoP(100).

3. Regarding the elemental map of FeP-NiMoP₂/FNF. It seems that P does not densify in the "core" part that is rich in Mo and Ni. This may suggest to the reader that the core structure is more of a NiMox alloy rather than a NiMoP₂ structure.

4. I suggest the authors to write down when their electric potential (voltage) is reported vs RHE as reference (eg, -1.0 V_RHE). Right now such a reference is stated very late in the manuscript, in line 450. This makes the reading confusing, as sometimes the authors seem to report the difference between anode and cathode potentials, and it is unclear what is the reference of most of the voltages.

Response to referees' comments

Reviewer #1

In the paper, Yang et al. reported Fe-Mo-based phosphide heterojunctions for both 4-NBA reduction and HMF oxidation. The catalysts were well-characterized and reaction pathways were well-investigated. However, these reactions on non-noble catalysts and their reaction pathways have been well-reported in the literature, such as, *ACS Sustainable Chemistry & Engineering* 9.5 (2021): 1970-1993, *CCS Chem.* 7, 507-515 (2020), *Angew. Chem. Int. Ed.* 58, 9155-9159 (2019). Considering the novelty of this work, it is hard to be accepted by Nature Communications.

Other comments are shown below.

Response: We greatly appreciate the reviewer for his/or her tireless review of our paper and thank the reviewer for the recognition on the characterization of our catalysts and the study of the reaction pathways. The novelty and significance of our work are further elucidated as follows:

About three literature as reviewer mentioned. We have read carefully and two of them were cited in our original manuscript. First two works (*ACS Sustainable Chemistry & Engineering* 9.5 (2021): 1970-1993; *CCS Chem.* 7, 507-515 (2020)) focus on the study of the 5-HMF EOR or 4-NBA EER half-reactions. While a feature of our work is the high-efficiency overall paired 5-HMF EOR and 4-NBA EER, which differentiates with above two. The third one (*Angew. Chem. Int. Ed.* 58, 9155-9159 (2019)) preliminarily investigated the paired electrocatalysis with NiB_x as both anode and cathode. However, the same material often exhibits difference in activity for two electrodes, resulting in a compromising performance for assembling overall electrolyzer. Particularly, the catalytic reaction mechanisms are not elucidated in these works. To expedite the advance of this nascent realm, pursuing more efficient electrocatalysts and clarifying the catalytic mechanism is highly important and meaningful.

In our work, to achieve synchronous and efficient paired electrocatalysis of 5-HMF EOR and 4-NBA EER, we have reasonably designed the matched electrocatalysts, where FeP-MoP supported on Fe foam (FF) for cathodic 4-NBA EER by learning from the biological [Fe-Mo]-nitrogenase and a similar material containing Ni (FeP-NiMoP₂/FNF) for catalyzing anodic 5-HMF EOR. The similar components of two electrodes are convenient for the industrial catalyst integration. Meanwhile, the introduction of Ni accelerates the reaction kinetics of anodic 5-HMF EOR, making the reaction kinetics of 4-NBA EER and 5-HMF EOR more matchable. The matchable design in these aspects contributes greatly to the impressive performance of our paired electrolyzer with low voltage expense, high faradaic efficiency and remarkable cycle stability. It reflects our rational and intellectual thinking in catalyst design for satisfying the highly efficient paired electrolysis of organic chemistry.

More importantly, we have systematically investigated the reaction mechanisms of the paired system, including identifying the active species and uncovering the origin of kinetic matchability for 4-NBA EER and 5-HMF EOR, which has rarely been

touched in previous reports. We corroborated water as sole H and O source rather than organics or dissolved oxygen, and clarified the specific water activation and dissociation and hydrogen/oxygen participation patterns. For anodic electrooxidation, the OH* species stemmed from the first step of water decomposition are determined as the active species. The first step of water decomposition requires the smallest energy compared with the subsequent three steps that produce OOH*, O* and O₂, thus avoiding the maximum energy barriers of water splitting. Meanwhile, for cathodic electroreduction, active H* species also come from the first step of water decomposition. Both the H and O species from the first step of water decomposition allow to the maximum matching of subsequent paired cathodic 4-NBA EER and anodic 5-HMF EOR with lowest energy consumption. Besides, water as sole H and O source for the paired system is a sustainable and recyclable route. Because the valuable target products can be continuously and efficiently produced by the addition of the reactants at two-electrode chambers without producing side products. Otherwise, if H or O species come from organics, such as H provided by hydroxyl groups of organics or dehydrogenation of anodic oxidation and O originated from deoxygenation of cathodic reduction, other reaction pathways occur and side reactions happen, resulting in reducing the selectivity and faradaic efficiency of target products. Or if dissolved oxygen from air as O source is also not sustainable due to the small amount. Above reaction mechanistic studies are greatly significant, which can not only guide for the development of this field and catalyst design, but also greatly extend to development and application of other fields with water as a feedstock.

Overall, the rational designability of our paired electrocatalysis and the deep mechanistic studies make that our work is novel and meaningful and worthy to be published in Nature Communications.

Q1. Line 11, the authors mentioned “active H* species for organic electroreduction originate from water.” Why did they want to highlight this in the abstract? Is there any other possible H source in the aqueous solution?

Response: We appreciate the reviewer for the comment. Actually, besides the H source for water, organics can probably provide the H source as described above, such as the hydroxyl groups of 4-NBA (*J. Am. Chem. Soc.* 138, 10128-10131 (2016)) or aldehyde dehydrogenation of anodic 5-HMF (*Nat Catal.* 5, 66-73 (2022)). By studying reaction mechanisms, we have verified that active H* species for organic electroreduction come from water. And only water as H (or O) source for the paired system is a sustainable and recyclable route. The relative discussion has been added in our revised manuscript (Paper 17-18, Line 423-432).

Q2. Line 28-29, “both hydrogen and oxygen evolution reactions (HER and OER) are slow kinetics”. We usually agree on that only OER is kinetically sluggish, which is major barrier to water electrolysis.

Response: We thank the reviewer for the comment. It may be that we did not describe it accurately. Comparatively, OER is more sluggish than HER owing to the complex four proton-coupled electron transfer as reviewer said. Following this comment, we

have rewritten this content as follow (Page 2, Line 28-31): “However, both hydrogen and oxygen evolution reactions (HER and OER) are inefficient due to the high activation barriers, and particularly the sluggish kinetic of complex OER process hinders the water electrolysis efficiency.”

Q3. Line 102-103, how did the authors test the nanosheet thickness by TEM? Why not just using the cross-section SEM to measure the thickness?

Response: We appreciate the reviewer for the constructive comment. Following this suggestion, we have carried out the cross-section SEM characterization to give the information of the nanosheet thickness. As shown, the FeP-MoP nanosheets with ~50 nm in thickness are clearly observed (inset of Fig. 1b). The relative content has been added in our revised manuscript (Paper 4, Line 106-109).

Fig. 1b | The SEM image for the cross-section of FeP-MoP nanosheet.

Q4. Line 136-137, should it be called magnetic FeNi₃ or alloy FeNi₃? It is not the metallic, right?

Response: Following the reviewer’s suggestion, we have revised it as alloy FeNi₃ in our manuscript (Paper 6, Line 135).

Q5. Line 161-162, “Such electron-rich Fe is similar to that of nitroreductase,²⁸ which is conducive to 4-NBA ERR.” The citation 28 is on nitrogen reduction to ammonia, not 4-NBA ERR. So what is the similarity of these two works and how transferable and comparable of the similar catalysts activity?

Response: We thank the reviewer for the comment. Two reactions have similarities in reaction mechanisms and both are intrinsically the cracking of nitrogen bonds (N=O or N≡N) and combination of nitrogen and proton (*Angew. Chem. Int. Ed.* 59, 12736-12740 (2020); *Angew. Chem. Int. Ed.* 59, 20411-20416 (2020)), which represent possible catalytically active commonality of 4-NBA ERR and NRR. Indeed, the active catalysts for NRR, such as Fe or Mo-based materials (*Chem. Rev.* 120, 5437-5516 (2020); *ACS Catal.* 10, 4914-4921 (2020)) also present large catalytic potential for 4-NBA ERR (*Chem. Rev.* 119, 2611-2680 (2019); *Angew. Chem. Int. Ed.* 58, 7794-7798 (2019)).

Q6. Line 174-176, “where FeP-MoP/FF exhibits a better 4-NBA ERR activity than FeP-NiMoP₂/FNF (Supplementary Fig. 9).” What could be the reason to explain

FeP-MoP/FF exhibits higher activity? Is it because of the higher surface area or any other possibilities? Did the authors test electrochemical active surface area (ECSA)?

Response: We thank the reviewer for the comment. Following this suggestion, the ECSA of FeP-MoP/FF and FeP-NiMoP₂/FNF were calculated (Supplementary Fig. 9c). FeP-MoP/FF shows the higher ECSA than FeP-NiMoP₂/FNF, indicating that the higher ECSA can contribute to part of good activity of FeP-MoP/FF but not all. More importantly, Fe site is identified as the active center for 4-NBA ERR. Electron accumulation on Fe active site facilitates to attract more protons to reduce -NO₂ into -NH₂. However, when introducing Ni, it traps electrons from Fe, verified by XPS result, resulting into the decrease of electrons on Fe site, which is not conducive for the electroreduction of 4-NBA. This is the intrinsic origin that the 4-NBA ERR activity of FeP-NiMoP₂/FNF is inferior to that of FeP-MoP/FF. Furthermore, the adsorption energy of 4-NBA (E_{4-NBA}) of FeP-MoP and FeP-NiMoP₂ was also calculated and compared (Supplementary Table 8). The higher E_{4-NBA} on FeP-MoP (-1.33 eV) than FeP-NiMoP₂ (-1.29 eV) further indicate the more favorable 4-NBA ERR kinetics on FeP-MoP surface. Above relative contents have been added in our revised manuscript (Paper 14-15, Line 349-354, SI: Paper 10, Line 89-91).

Supplementary Fig. 9 | (c) Capacitive curves of FeP-MoP/FF and FeP-NiMoP₂/FNF in 1.0 M KOH with 10 mM 4-NBA.

Supplementary Table 8 | The adsorption energies of 4-NBA on different sites of FeP-MoP, FeP-NiMoP₂, FeP and MoP surfaces.

Models	FeP-MoP	FeP-NiMoP ₂
Energy (eV)	-1.33 (Fe site)	-1.29 (Fe site)
	-1.08 (Mo site)	-1.12 (Mo site)
	-0.91 (P-FeP site)	-0.96 (Ni site)
	-0.81 (P-MoP site)	-0.90 (P-FeP site)
		-0.75 (P-NiMoP ₂ site)

Q7. Figure 3b-c, Figure 3h, and line 197-199. The authors tested the LSV and ECSA among three catalysts. It seems that the higher activity of FeP-MoP/FF than FeP/FF and MoP/FF is mainly originated from the higher surface area of FeP-MoP/FF? What is the ECSA-normalized reaction activity (or current density)?

Response: We thank the reviewer for the constructive comments. Following this comment, we have supplied the ECSA-normalized activity of FeP-MoP/FF, FeP/FF, and MoP/FF catalysts (Supplementary Fig. 13). As shown, FeP-MoP/FF delivers higher ECSA-normalized activity compared to the FeP/FF and MoP/FF, further suggesting the excellent intrinsic 4-NBA ERR activity of FeP-MoP heterojunction. Actually, apart from the high surface area, other merits of FeP-MoP/FF, such as highly active heterointerface and rapid electronic conductivity also contribute the enhanced activity. The excellent activity of FeP-MoP/FF is the result of comprehensive effects. The relative content has been added in our revised manuscript (Paper 9, Line 207-209, SI: Paper 14, Line 115-120).

Supplementary Fig. 13 | LSV curves with normalization by ECSA of FeP-MoP/FF, FeP/FF and MoP/FF

Q8. Line 203, “the electroreduction of 4-NBA was carried out at the potential of 0.004 V”. Is it 0.004 V vs. RHE? Why did the authors select so positive potential for electroreduction? Please check this value.

Response: We thank the reviewer for the comment. Yes, the potential of 0.004 V is versus the RHE potential. In our electrochemical test, the Hg/HgO electrode was used as reference electrode. Then the potentials were quoted with respect to the reversible hydrogen electrode through $E_{RHE} = E_{HgO/Hg} + 0.059 \times pH + 0.098$ V. The reason to select the potential of 0.004 V vs. RHE is to avoid the HER interference (Since HER initiates after this value and it overlaps with 4-NBA ERR). The positive potential applied for electroreduction is commonly reported in the literature (e.g. *ACS Catal.* 11, 13510-13518 (2021)). Above content has been added in our revised manuscript (Paper 8, Line 178-180).

Q9. Similar questions as #7 for FeP-NiMoP₂/FNF to catalyze 5-HMF EOR.

Response: According to this constructive suggestion, the ECSA-normalized activity

of catalysts for 5-HMF EOR are provided (Supplementary Fig. 22). Similarly, FeP-NiMoP₂/FNF also delivers higher activity compared to the FeP-MoP/FF, NiMoP₂/NF and FeP/FF, further suggesting the superb intrinsic 5-HMF EOR activity of FeP-NiMoP₂/FNF heterojunction. The relative content has been added in our revised manuscript (Paper 10, Line 251-252 and Paper 11, Line 261-262 SI: Paper 23, Line 204-208).

Supplementary Fig. 22 | Polarization curves with normalization by ECSA of FeP-NiMoP₂/FNF, NiMoP₂/NF, FeP-MoP/FF and FeP/FF

Q10. Line 352-353, DFT results showed that “Based on charge analysis, the electrons of Mo transfer to Fe (1.36 e⁻) and Ni (0.88 e⁻), respectively.” This conflicts with the XPS results (line 167), Ni captures electrons from Fe and Mo. How to explain this inconsistency?

Response: Thank the reviewer very much for the comment. DFT results and XPS results are consistent and not contradictory. Mo site is the active center in FeP-NiMoP₂ heterojunction for 5-HMF electrooxidation. From the point of active center, trapping electrons from Mo is favorable for the electrooxidation of 5-HMF. DFT results show that the electrons of Mo transfer to Fe (1.36 e⁻) and Ni (0.88 e⁻), which means that Fe and Ni both have the ability to trap electrons from Mo. Meanwhile, XPS results exhibit that Ni captures electrons from Fe and Mo, which indicates that Ni has more stronger capturing electron ability than Fe, which not only captures electrons from Mo, but also captures electrons from Fe. Therefore, the XPS and DFT results are consistent. This is also the reason that we design the introduce of Ni in Fe-Mo-based composite, which can improve the reaction kinetics and activity of 5-HMF EOR. The relative discussion has been added in our revised manuscript (Paper 7, Line 171-175).

Q11. Line 367-379, the authors used isotopic labelling and confirmed the proton and oxygen source are both from water. Are there any other possibilities of proton/oxygen sources? Based on previous literatures and the high conversion and selectivity in this work, the H/O source would not be coming from the contaminations, it is obviously from H₂O to provide H/O source.

Response: Thank the reviewer very much for the comment. For cathodic reaction, besides the H source from water, H source may come from the hydroxyl groups of 4-NBA and aldehyde dehydrogenation of anodic 5-HMF as responded in Q1. With respect to anodic reaction, oxygen source may come from the deoxygenation of 4-NBA cathodic reduction, including the nitro deoxygenation and hydroxyl deoxygenation (*ACS Catal.* 9, 8068-8072 (2019); *Energy Environ. Sci.* 13, 917-927 (2020)) or dissolved oxygen originated from air (*Green Chem.* 14, 143-147 (2012)) in addition to that from water. Just due to the H/O source from water for the paired electrocatalysis of organics, no other side products producing, so such high conversion and selectivity of catalysts were achieved. Thus the systematical and deep investigation of the catalytic reaction mechanisms, including confirming the proton and oxygen source by using various means, such as isotopic labelling, is very important and meaningful. The relative content has been added in our revised manuscript (Paper 17-18, Line 423-432).

Q12. HMF is not stable in strong alkaline solutions, e.g., 1 M KOH, *ACS Catal.* 2018, 8, 2, 1197-1206. Did the authors observe the degradation of HMF or Cannizzaro reaction in the electrolyte?

Response: We appreciate the reviewer for the valuable comment. The Cannizzaro reaction in the alkaline solution is an endothermic reaction. The reaction rate is very slow at low or normal temperature and it takes a long time. The same is for degradation reaction. We further performed some additional experiments to verify this point (Supplementary Fig. 18). As shown, there is no side products and the HMF concentration change in 1.0 M KOH solution (pH 14) containing 10 mM HMF at the temperature of 298K (test conditions in our experiment) for a long time (24 h) with no adding catalysts, suggesting that HMF is stable in strong alkaline solution without the degradation of HMF or Cannizzaro reaction. When adding our efficient catalyst, the time of converting HMF (10 mM) is very short (<2 h), so none of these reactions occur in our experiment. When increasing the concentration of HMF to 100 mM, the time of converting HMF (100 mM) is relative short (<5 h) by using our catalyst, along with high conversion and selectivity ($\geq 99\%$ and $\geq 98.0\%$). During this period still no the degradation of HMF or Cannizzaro reaction was observed. Additionally, considering industrial scale-up application, a large volumes of more higher concentrated HMF solutions (>0.5M) are usually used and HMF remains in solution for extended periods of time. There may occur the the degradation of HMF or Cannizzaro reaction in strong alkaline solutions (pH 14) as reviewer said and the literature reported (*ACS Catal.* 8, 2, 1197-1206 (2018)). So the pH value of 14 may not be a viable condition for industrial processes. We also notice this point. In our work, weak alkaline or even neutral systems (pH= 11.63, 10.02, 8.31 and 6.86) were explored. FeP-NiMoP₂/FNF shows excellent activities in a wide pH range for HMF EOR, implying the good universality of our catalyst. Our further studies will pay more attention to the exploration of this reaction in weak alkaline or neutral systems. The relative content has been added in our revised manuscript (Paper 10, Line 235-236 SI: Paper 19, Line 168-172).

Supplementary Fig. 18 | (a) HPLC-trace acquired at various times and (b) the corresponding stability test of 5-HMF in 1.0 M KOH containing 10 mM HMF under string at the temperature of 298K for 24 h. The chronoamperometric response curve of FeP-NiMoP₂/FNF in 1.0 M KOH containing (c) 10 mM and (d) 100 mM HMF. (e) HPLC-trace acquired at various times and (f) the corresponding stability test of 5-HMF in 1.0 M KOH containing 100 mM HMF under string at the temperature of 298K for 6 h.

Q13. Figure S15 and S22. How did the authors prepare different pH electrolytes? Did the authors use buffer solutions? If not, the electrolyte would quickly change to alkaline/acidic solution during ERR (proton consumption reaction) and EOR (proton generation reaction) reactions, respectively. So that the pH of electrolytes before and after reactions would show large differences. Therefore, the calculation of potential versus RHE and explanation to reaction mechanisms may have problems.

Response: We are thankful for the reviewer's comment. In our experiment, the different pH electrolytes were prepared by using buffer solutions, in which the K₂HPO₄-K₃PO₄ buffer solution for pH=11.63, K₂CO₃-KHCO₃ buffer solutions for pH=10.02 and 8.31 and K₂HPO₄-KH₂PO₄ buffer solution for pH=6.86. The pH values of electrolytes did not obviously change during ERR and EOR processes. The specific description has been added in our revised manuscript (Page 20, Line 511-513).

Reviewer #2

The authors have designed a paired electrolysis system using Fe-Mo based electrodes for simultaneously reduction and oxidation of organics with water as the feedstocks, which can be driven by a solar cell with extremely low electrolytic voltages and high Faradaic efficiency. This work will be highly interested for the researchers in electrocatalysis, electrosynthesis, green chemistry, and energy conversion. In addition, this manuscript is well organized and well presented. Thus, this interesting manuscript deserves being published in Nature Communications after addressing the following minor concerns.

Response: We genuinely appreciate the reviewer for very positive evaluation of our work and for recommending our work for publication.

Q1. The sentence “Ni based material serve as the catalysts for 5-HMF EOR” on line 68-69 stands out here, which is logically incoherent with the context. The authors is encouraged to rewrite this part to expand the elaboration.

Response: We appreciate the reviewer for the constructive suggestion. According this suggestion, we have revised original manuscript to further elaborate the important role of nickel (Ni) on significantly elevating 5-HMF EOR activity of catalysts in the introduction part as follows (Page 3, Line 71-74): “Ni is considered as ideal candidate, in view of the fact that the multivalent characteristic of Ni making it promising for organic oxidation^{34,35}. So the introduction of Ni into the Mo-Fe system is expected to improve the selectivity and stability of HMF EOR.”

Q2. In Fig. 3b, the yellow line emerges a sharp current increase at about 0.3 V, making the turning point presents a strange, uneven trend like an obtuse angle. Similarly, it is also observed on the green line in supplementary Fig. 11. It's rarely seen in the LSV test. It means some oxidation peaks? Please explain them briefly or provide new data.

Response: We thank the reviewer for the constructive suggestion. We sincerely apologize for this carelessness. The peak at about 0.3 V is assigned to the oxidation peak of Fe from Fe foam substrate[*Lide, D. R. et al. CRC Handbook of Chemistry and Physics: 86th Edition. (CRC Press, Boca Raton, 2006)*]. It may be that our coating or preparing the samples is not careful enough, leading to the nudity of part of Fe substrate, thus producing such interference. Following this suggestion, we have re-prepared the samples and re-performed the experiments with three times. The revised data was provided as Fig. 3b and supplementary Fig. 11 in our revised manuscript.

Fig. 3b | LSV curves of FeP-MoP/FF, FeP/FF and MoP/FF.

Supplementary Fig. 11 | LSV curves of synthesized FeP-MoP/FF by using different APM concentrations for 4-NBA ERR.

Q3. The statement “Only Sun’s group ...” on line 58 is inaccurate. Actually, paired electrolysis strategy has been applied in some early works (e.g. Ref. 25; doi: 10.1002/anie.202009155; doi: 10.1002/anie.202009757). Thus, it is suggested to revise the related description. The author list in Ref. 25 is wrong. Please double check the author lists of Ref.

Response: We thank the reviewer for the constructive suggestion. We have revised the related description in the revised manuscript (Paper 3, Line 58-64) as “Sun’s group preliminarily investigated the paired electrocatalysis of organics with NiBx as the catalyst²⁷, but with relatively high energy consumption. Other derivative systems, such as cathodic electrocatalytic deuteration and its paired transformations with anodic amine or alcohol oxidation, have also been studied recently^{28,29}.”

In addition, we are sorry for this error. Ref. 25 was corrected and the excellent and highly related work (doi: 10.1002/anie.202009155; doi: 10.1002/anie.202009757) has been added as Ref. 28 and Ref. 29 in the revised manuscript.

Q4. All the potentials normalized to RHE may be inaccurate because the pH value often vary during the water-involving organic reaction. Generally, all the electrochemical data vs the actual reference electrode should be provided in the text (see the ref. Nature Communications 2021, 12, 3881). So, the data vs Hg/HgO are suggested in the manuscript, or the potential vs Hg/HgO electrode should be add to Fig. 3a, the fist electrochemical figure.

Response: We thank the reviewer for the valuable suggestion. The potential vs Hg/HgO electrode was added on top axis in Fig. 3a and other electrochemical figures and the highly related work (*Nat. Commun.* 12, 3881 (2021)) had been added as Ref. 43 in our manuscript.

Fig. 3a | LSV curves of FeP-MoP/FF in 1.0 M KOH without and with 10 mM 4-NBA at a scan rate of 5 mV s⁻¹.

Q5. In Fig. 5a, how is the voltage of the two-electrode electrolytic system converted into the relative RHE potential? Here I think it is the cell voltage. "Potential (V vs. RHE) " should be correct to be "Cell voltage (V vs. counter electrode)", "E (V vs. counter electrode)" or "E (V)".

Response: We sincerely apologize for this error. It is the "E (V)", which has been revised in our manuscript. Thanks again.

Fig. 5a | LSV curves of FeP-MoP/FF||FeP-NiMoP₂/FNF couple with and without organic substances.

Reviewer #3

Yang and coworkers presented a family of electrocatalysts that reduce/oxidize organic molecules with very high faradaic efficiencies, compared to the background hydrogen and oxygen evolution reactions (HER/OER). FeP-MoP supported on Fe foam (FF) and a similar material containing Ni (FeP-NiMoP₂/FNF) were the best performers for reduction of 4-NBA and oxidation of 5-HMF respectively. The combined oxidation/reduction of such organic molecules demonstrate an appealing path to be further explored. Before publication, however, there are blind spots that need clarification.

Response: We genuinely appreciate the reviewer for very positive evaluation of our work and thank the reviewer for constructive comments, which help to improve the quality of our article. The questions raised by the reviewer have been fully addressed.

Q1. The main one I currently observe is related to the models in Suppl Fig 26, specifically FeP-NiMoP₂ (e). It seems that the authors produced it from that of FeP-MoP (d). The specific details about how the authors produced both of them should be dully explained. For instance, it seems that they commensurated FeP and MoP in a 3:2 ratio (Figure 1d), but that is not said anywhere (eg, lines 107-108 and 120 do not say that). Same with NiMoP₂ (Figure 1h).

Response: We thank the reviewer for the valuable comment. Actually, the FeP-NiMoP₂ model (Suppl Fig. 26e) was built based on single FeP (Suppl Fig. 26a) and NiMoP₂ (Suppl Fig. 26c) phases, instead of that of FeP-MoP. Additionally, the crystal planes of theoretical modeling for NiMoP₂, FeP and MoP are based on the exposed planes of their TEM results, where FeP and MoP or NiMoP₂ were matched well when the FeP and MoP or NiMoP₂ were commensurated in a 3:2 ratio. Therefore, the 3×2 FeP supercell, 2×2 MoP and 2×2 NiMoP₂ were used as crystal models for facet cleavage and construction of heterojunctions and a four-layer of slab supercell was chosen as the surface slab supercell to decrease the complexity of calculation. Following this constructive suggestions, we have added the specific description about the model establishment of FeP-NiMoP₂ and FeP-MoP in the manuscript (Page 22, Line 543-554).

Q2. Can these surfaces reconstruct upon cleavage? NiMoP₂ (001) and MoP(100).

Response: We thank the reviewer for the excellent comment. Generally, when cutting the crystal, it is metastable (due to high energy) relative to the bulk phase, surface reconstructions (or defects) are produced to reduce the energy. Due to the complexities of surface reconstructions, including numerous configurations, fully investigating this effect is difficult for both experimentally and theoretically. The premise that surface reconstruction affects the heterostructure is the heterojunction synthesized by a two-step method, which one phase is synthesized firstly, and then another phase is synthesized on the basis of the previous phase and the surfaces of both two phases will reconstruct to form a heterojunction. Our FeP-MoP and FeP-NiMoP₂ heterojunctions are synthesized by one-pot method. The Fe-Mo oxide or

Fe-NiMo oxide precursors are fabricated by synchronously adding the metal sources, in which the Mo, Ni and Fe metals in the formed precursors are fully coordinated and have the lowest free energy. By further converting into phosphide heterojunctions, it does not involve the issue of surface reconstructions of two surfaces.

Q3. Regarding the elemental map of FeP-NiMoP₂/FNF. It seems that P does not densify in the "core" part that is rich in Mo and Ni. This may suggest to the reader that the core structure is more of a NiMox alloy rather than a NiMoP₂ structure.

Response: We thank the reviewer for the comment. Based on XRD (Fig. 2b), there is the existence of NiMoP₂ phase, but no the phase of NiMo_x alloy in the heterojunction, indicating the structure of the composite is NiMo phosphide rather than NiMo_x alloy. The relative low density of P can be due to low amount of P than Mo and Ni, which is verified by XPS results (Supplementary Table 7). In addition, generally, in the elemental mapping images, the elements with larger atomic numbers, such as Mo, Ni, looked brighter than C, P, and Si that have smaller atomic numbers (*Nanoscale*, 12, 16586-16595 (2020)). This may be an other reason that P looks like low dense than Mo and Ni.

Fig. 2b | The XRD patterns of FeP-NiMoP₂/FNF.

Supplementary Table 7 | XPS analysis of FeP-NiMoP₂/FNF

Element	Fe	Ni	Mo	P
XPS (wt.%)	25.21	32.32	30.02	12.45

Q4. I suggest the authors to write down when their electric potential (voltage) is reported vs RHE as reference (eg, -1.0 V_RHE). Right now such a reference is stated very late in the manuscript, in line 450. This makes the reading confusing, as sometimes the authors seem to report the difference between anode and cathode potentials, and it is unclear what is the reference of most of the voltages.

Response: We thank the reviewer for the constructive suggestion. Following this suggestion, the description about potential (voltage) vs. RHE is added in our revised manuscript. Thanks again.

REVIEWERS' COMMENTS

Reviewer #1 (Remarks to the Author):

The authors have addressed the comments. I agree to accept it for its publication in Nature Communication without further revisions.

Reviewer #2 (Remarks to the Author):

Fu et al. have carefully addressed all the concerns of our three reviewers and provided a satisfactory response/revision.

Paired electroreduction and electrooxidation of organics with water as a feedstock to produce value-added chemicals is now in an emerging and highly active field. This rationally designed work provided a deep insight into the mechanism of the paired electroreduction and electrooxidation of organics on Fe-Mo-based phosphide heterojunctions by various characterization results and detailed discussions. This paired electrosynthesis driven by a solar cell with extremely low electrolytic voltages and high Faradaic efficiency highlighted the promising potential. I believe that the interesting results are solid and its novelty fit our high expectation of Nature Communications.

Thus, I would like to highly recommend the publication of this work in Nature Communications at its present version.

Reviewer #3 (Remarks to the Author):

The work of Yang and coworkers is an excellent fit for Nature Communications: It is clearly written, the mechanism has been robustly elucidated and opens new avenues to be explored, and the modelling part is well above standards. I hereby recommend publication in its current form.